# Gamma-ray flares from relativistic magnetic reconnection in the jet of the quasar 3C 279

A. Shukla [1,2✉] & K. Mannheim[1]

Spinning black holes in the centres of galaxies can release powerful magnetised jets. When the jets are observed at angles of less than a few degrees to the line-of-sight, they are called blazars, showing variable non-thermal emission across the electromagnetic spectrum from radio waves to gamma rays. It is commonly believed that shock waves are responsible for this dissipation of jet energy. Here we show that gamma-ray observations of the blazar 3C 279 with the space-borne telescope Fermi-LAT reveal a characteristic peak-in-peak variability pattern on time scales of minutes expected if the particle acceleration is instead due to relativistic magnetic reconnection. The absence of gamma-ray pair attenuation shows that particle acceleration takes place at a distance of ten thousand gravitational radii from the black hole where the fluid dynamical kink instability drives plasma turbulence.

[1] Institut für Theoretische Physik und Astrophysik, Universität Würzburg, Emil-Fischer-Str. 31, 97074 Würzburg, Germany. [2] Discipline of Astronomy, Astrophysics and Space Engineering, Indian Institute of Technology Indore, Khandwa Road, Simrol, Indore 453552, India. ✉email: amit.shukla@iiti.ac.in

Accreting black holes are suspected to convert rotational energy into Poynting flux escaping along their rotation axes and powering collimated jets[1,2]. The recent Event Horizon Telescope (EHT)[3] observation of the active galactic nucleus (AGN) M 87 provides corroborating evidence for this scenario[4]. It is not known yet how and where the energy of the twisted magnetic fields near the black hole is converted into the kinetic energy of the particles in the jet[5].

Blazars are commonly understood as jets emerging from AGN viewed under small angles with the line of sight. The relativistic motion of the plasma boosts the non-thermal jet emission into a forward cone. It is commonly believed that the emission is produced by ultra-relativistic particles accelerated at shock waves travelling down the jet[6]. Shock acceleration, however, may not be efficient in the magnetically dominated relativistic plasma[7,8]. The time-variability of the emission at the highest frequencies allows to witness the physical processes going on in the innermost parts of the jets[9]. Recent detections of minute-scale gamma-ray variability in blazar light curves imply limits on the sizes of the high-energy emission region that challenge the shock-in-jet scenario[9–11]. Denoting the variability time-scale as $\Delta t = 60 \Delta t_{min}$ s, and assuming incoherent emission from a region of size $r$ moving with bulk Lorentz factor $\Gamma_j = 35 \Gamma_{35}$ at a cosmological redshift $z$, causality requires that in units of the Schwarzschild radius $r_S = 2GM/c^2$ of a black hole with mass $M$, the region must be smaller than $r/r_S = 2.1 m_8^{-1} \Gamma_{35} \Delta t_{min} (1 + z)^{-1}$. In blazars, the black hole masses are typically very large such that $m_8 = M/10^8 M_\odot > 1$ where $M_\odot$ denotes one solar mass, i.e. the gamma-ray emitting regions must be smaller than the black hole horizon. By comparison, a jet collimated to the light-cylinder radius will develop shock waves with a radius exceeding $10 r_S$[12].

Moreover, the recent global magnetohydrodynamic and three-dimensional particle-in-cell simulations showed that the kink instability can disrupt the jet[13,14], and can lead to efficient particle energisation[15]. Brightening of the jets is expected to occur as the jet becomes unstable to current-driven kink or shear-flow instabilities producing a filamentary current density pattern prone to relativistic magnetic reconnection[16]. In this low-density region, the magnetic field energy dissipates mainly by particle acceleration and non-thermal emission. The particles could be electron-positron pairs created at the base of the jet by photon-photon collisions or particles entrained from the ambient medium. Electrons and positrons produce gamma-rays by inverse Compton scattering[17]. Protons and ions produce pions in collisions with low-energy photons which decay into gamma-rays and high-energy neutrinos[18,19]. Surprisingly, in blazars of the flat-spectrum radio quasar (FSRQ) type exhibiting broad line regions (BLR), the gamma-rays from the compact emission regions must originate lightyears away from the central black hole: no sign of gamma-ray attenuation due to pair production from collisions between the gamma-rays and the UV photons from the BLR is generally found[9–11].

3C 279 is a blazar of FSRQ type with a black hole mass of $(3-8) \times 10^8 M_\odot$[20,21] at a distance corresponding to 5 billion years of light travel time. The source emits gamma-rays up to 400 GeV[22], and its flux is known to vary on short time-scales[11,23–25]. The blazar 3C 279 is known for extreme gamma-ray variability[26]. A giant flare in 2015 showed hourly time scale flux variations occuring outside of the quasar's BLR at a distance of 0.05 parsec estimated for the formation of dissipative shock waves. During the Fermi-LAT era, several strong outbursts[26–28], including the one of June 2015 were observed. This outburst showed flux doubling in just a few minutes reaching an isotropic luminosity of $10^{49}$ erg s$^{-1}$ in its maximum[11].

Here, we report the analysis of a similar giant flare in 2018, which shows minute-scale variability superimposed on the longer duration flare. The shock dissipation scenario may not be appropriate for the interpretation, but instead magnetic reconnection might play the dominant role for the dissipation of the jet energy in this stage of its dynamical evolution. This peak-in-peak variability pattern originates from structures much smaller than shocks yet at a large distance from the central machine. Outside of the BLR, the collimated jet is expected to become unstable, developing a medley of reconnecting flux ropes where Poynting flux is converted into kinetic-energy flux. We interpret a minute-scale-flare superimposed on a longer duration flare envelope as the signature of co-spatial emission due to relativistic magnetic reconnection from this transition region.

## Result

**Sequel of identical flares and co-spatial emission**. Fermi-LAT is a pair-conversion detector covering the energy range from about 20 MeV to more than 500 GeV[29]. It has been operated primarily in an all-sky survey mode, for details see "Methods". 3C 279 showed enhanced activity across the entire electromagnetic spectrum from January to June 2018[30,31]. In this period, three pronounced gamma-ray flares F1 (MJD 58133 - MJD 58139), F2 (MJD 58222 - MJD 58232), and F3 (MJD 58268 - MJD 58276) with durations corresponding to time-scales of days were detected (see Fig. 1a). The average gamma-ray spectra of all three flares, during the aforementioned period, were obtained by fitting a log parabola model and are represented in Fig. 1c.

The detection of high-energy photons above 13 GeV during these flares F1, F2, and F3 shows that the location of gamma-ray emission must be outside of the BLR to avoid pair absorption in collisions with low-energy photons[32]. Out of the three flares, F1 and F2 showed a peak-in-peak light curve, where fast flares were superimposed on the more slowly varying envelope emission. In all three flares, the spectral and timing properties of the envelope emission were found to be identical, suggesting a common emission process, similar size of the emission region, and a common origin. Notably, the rise and decay times of the fast flares are also identical, suggesting their common origin (see Table 1).

During flare F2, the similarity of the spectral curvature of the fast flares with the envelope emission strongly suggests that the fast flares and envelope emission might have originated from the same location in the jet, suffering exposure to the same external target photon fields for inverse-Compton scattering by electrons or photo-pion production by protons.

**Peak-in-peak light curve and minute-scale flare**. A detailed light curve of F2, one of the brightest flares ever recorded using Fermi-LAT from any AGN, is plotted in Fig. 1b. On 19 April 2018, Fermi-LAT detected two ultra-bright fast flares FF1 and FF2 on top of a slowly varying envelope of the flare F2, see Fig. 1b. Orbit-size binned light curves (96 min) of both the fast flares are plotted in maroon. Further, the slowly varying envelope component of F2, with 3-hour bins, is plotted in blue. A significant (4.7 σ) 8-min (or probably shorter) flare was superimposed on a longer duration envelope at the starting of the first fast flare FF1. The time measurement relates to a single orbit, thereby leaving no doubt about the reality of the phenomenon, see Fig. 2a. The average spectrum during this orbit was found to be very hard with photon index $\alpha = 1.74 \pm 0.08$. Fermi-LAT also observed one of the so-far greatest flux from the source (above 100 MeV) during one single orbit $(3.4 \pm 0.2) \times 10^{-5}$ ph cm$^{-2}$ s$^{-1}$ of the second fast flare of F2. A low-amplitude fast flare was also noticed just after the two bright fast flares during the envelope emission.

The average spectral energy distribution of the first two bright fast flares and envelope emission observed during F2 are

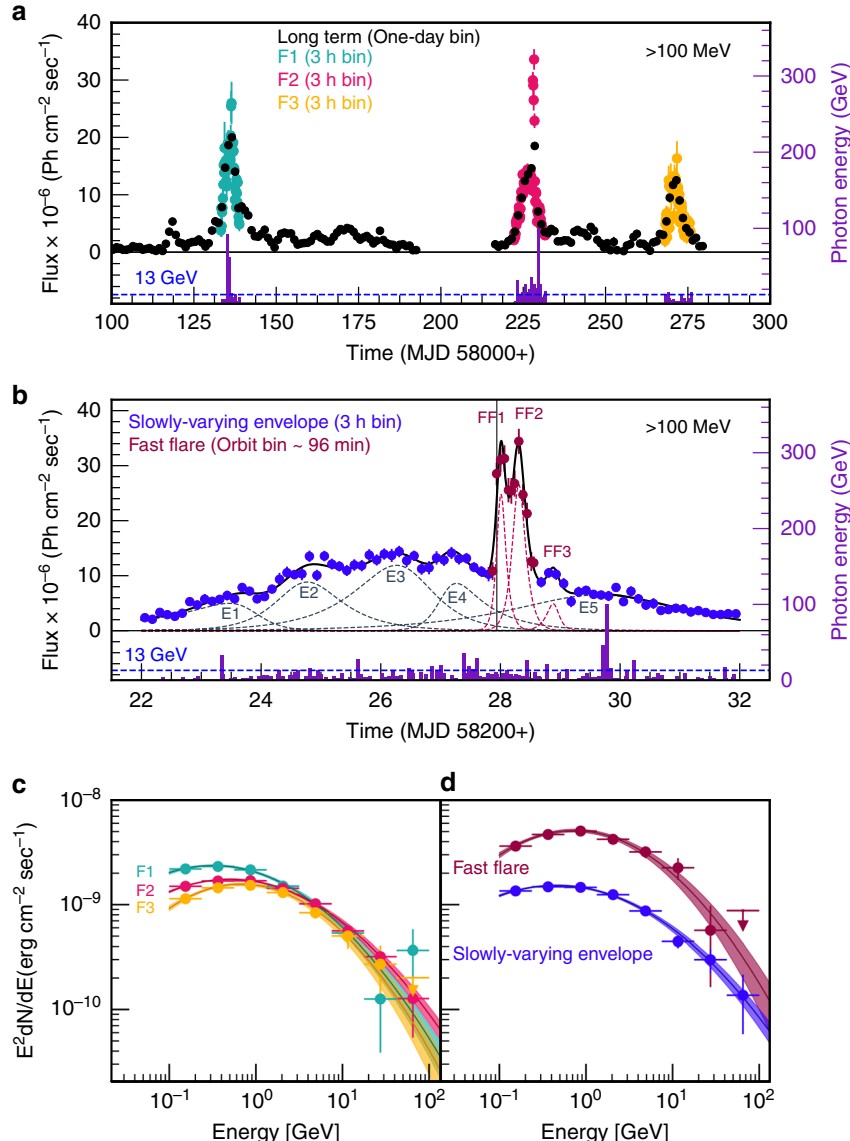

**Fig. 1 Gamma-ray flare characteristics.** Long-term light curve and three pronounced flares (F1, F2, F3) during MJD 58100 to MJD 58280 from 3C 279 are shown in **a** and a magnified view of F2 is plotted in **b** (left Y-axis). A peak-in-peak light curve, previously unknown to flare activity, is significantly observed during F2 where two fast flares are superimposed on the slowly-varying envelope emission. The energy of each observed photon (right Y-axis) with 99.99% or higher probability of association with 3C 279, is shown as violet bars for all flares in **a** and for F2, in **b**. Gamma-ray photons above 13 GeV are expected to be absorbed by the UV photons in the BLR due to a high optical depth for pair creation in photon-photon collisions[32]. Hence, the detection of photons with energy above 13 GeV during flares, strongly suggests that gamma-ray emission must have originated outside of the BLR. The spectral energy distributions (SED) of F1, F2, and F3 are plotted in **c** whereas **d** represents the SEDs of the slowly varying envelope (blue) and fast flare component (maroon) observed during F2. The error bars in the light curve and SEDs represent 1 $\sigma$ uncertainty.

presented in Fig. 1d. The highest-energy photon of ~100 GeV was observed during the envelope emission phase of F2. A ~27 GeV photon was detected during the decay of the second fast flare FF2, see Fig. 1b.

## Discussion

The minute-scale variability during the fast flare superimposed on the slowly varying envelope emission within flare F2, together with the spectral information showing the absence of pair absorption, constrains the size and location of the emission region. The observed peak-in-peak light curve can not be explained within the shock-in-jet scenario as the variability time scale would have to be larger than the light crossing time across the jet radius which itself exceeds the Schwarzschild radius[33]. The shock-in-jet scenario also has difficulties producing the similar

observed spectral curvatures of the envelope and fast flare. Moreover, differential Doppler boosting[34] has difficulties to explain the observed time profiles of the envelope and fast flare emission simultaneously with the spectral curvature.

By contrast, the jet-in-jet model[35] predicts mini-jets from magnetic reconnection events that result in a peak-in-peak structure matching the observed gamma-ray light curve and its spectral shape. Magnetic reconnection is triggered by instabilities disrupting the collimated jet flow[36]. The current-driven kink instability induced by non-axisymmetric perturbations could play a leading role in this context, and explain the helical trajectories of Very Large Baseline Interferometry (VLBI) components. In FSRQs, jet collimation is supported by a powerful wind from the accretion disk, preventing instabilities to develop inside of the BLR[37]. After the onset of instability outside of the BLR, turbulence develops leading

**Table 1 Temporal characteristics of Flares.**

| Flare (MJD) | Components [Name] | $T_r$(days) | $T_d$(days) | Reduced-$\chi^2$(DOF) |
|---|---|---|---|---|
| F1 (58133.0–58139.0) | Envelope [1] | 1.62 ± 0.06 | 1.49 ± 0.05 | 1.77 (43) |
| (Fitted with 2 components) | Fast flare [1] | 0.06 ± 0.02 | 0.04 ± 0.02 | |
| F2 (58222.0–58232.0) | Envelope [1] | 1.61 ± 0.03 | 2.74 ± 0.05 | 1.52 (79) |
| (Fitted with 3 components) | Fast flare [1] | 0.05 ± 0.01 | 0.09 ± 0.02 | |
| | Fast flare [2] | 0.09 ± 0.02 | 0.08 ± 0.01 | |
| F3 (58268.0–58276.0) | Envelope [1] | 1.47 ± 0.08 | 1.88 ± 0.06 | 1.69 (61) |
| (Fitted with 1 component) | | | | |
| F2 (58222.0–58232.0) | Envelope [E1] | 1.25 ± 0.17 | 0.25 ± 0.06 | 1.14 (65) |
| (Fitted with 8 components) | Envelope [E2] | 0.42 ± 0.05 | 0.57 ± 0.05 | |
| | Envelope [E3] | 0.75 ± 0.11 | 0.46 ± 0.17 | |
| | Envelope [E4] | 0.18 ± 0.07 | 0.59 ± 0.09 | |
| | Envelope [E5] | 1.35 ± 0.34 | 1.24 ± 0.08 | |
| | Fast flare [FF1] | 0.06 ± 0.01 | 0.08 ± 0.01 | |
| | Fast flare [FF2] | 0.09 ± 0.01 | 0.10 ± 0.01 | |
| | Fast flare [FF3] | 0.11 ± 0.04 | 0.08 ± 0.03 | |

Column (1) presents the name of the flare and their duration, next row in column (1) presents number of fitted components for the same flare, Column (2) presents the type of components used for fitting and their names, Column (3) presents the rise time of the flare, Column (4) presents the decay time of the flare, Column (5) presents the reduced-$\chi^2$ and degrees of freedom (DOF) of the overall fit for the flare with all the components.

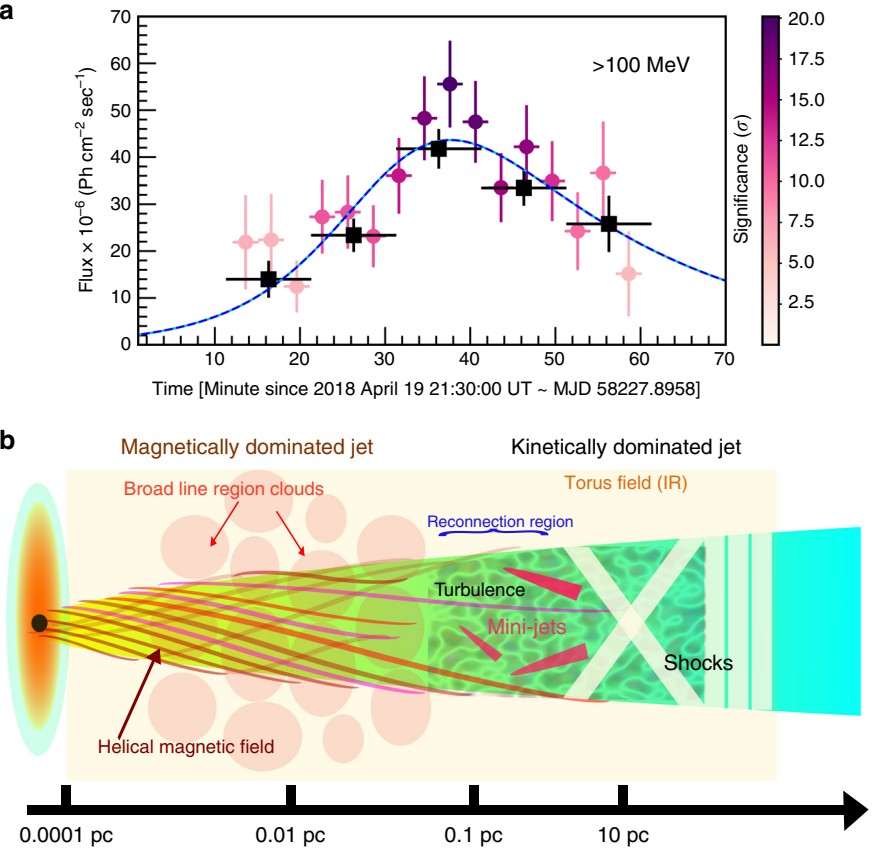

**Fig. 2 Minute-scale flare as a diagnostic tool for the jet geometry. a** Shows the 3-min (circle) and ten-minute (square) binned light curves measured during the orbit wherein a strong rapid variability of the order of a few minutes with high significance (4.7 σ) was observed. During this orbit, 3C 279 was found to be highly inconsistent with the constant flux having p-values 0.002 ($\chi^2$-test) and $10^{-5}$ ($\chi^2$-test) for the three-minute and 10-min binned light curves, respectively. Hence, the best-fit function to the 3-min binned light curve is deduced using the sum of two exponentials represented by the dashed cyan-blue line. Notably, the error bars in the light curve represent 1 σ uncertainty and the significance of each 3-min bin is colour-coded. **b** Shows a proposed sketch of the inner jet of a blazar, explaining the peak-in-peak light curve with reference to the jet-in-jet magnetic reconnection model. In this scenario, the magnetic field fragments into small plasmoids that interact and grow into monster plasmoids within the reconnection region. Subsequently, these massive plasmoids lead to the formation of mini-jets, which produce optically thin minute-scale gamma-ray flares. These mini-jets are represented in the sketch in magenta. Moreover, the emission from the reconnection region, as a whole, accounts for the observed envelope emission.

to the formation of a medley of twisted flux tubes with filamentary current sheets subject to further instabilities and the emergence of plasmoids. This is the reconnection region in which some plasmoids can grow to a sizeable fraction of 10% ($f = 0.1$) of the reconnection region through mergers before leaving the region and becoming monster plasmoids[35].

Here we apply the jet-in-jet model to explain the observed peak-in-peak light curve behaviour observed during the flare F2, where a minute-scale-flare was superimposed on a slowly varying envelope with a rise time of $t_{env} \sim 1.2 \times 10^5$ s. The characteristic size of the magnetic reconnection region in jet's co-moving frame can be expressed as $l' = t_{env}\Gamma_j\epsilon c/(1+z)$, where $\epsilon$ parameterises the reconnection speed. Further, by considering $\epsilon \sim 0.1$ as a typical observational inferred value and adopting a Lorentz factor of the jet $\Gamma_j \sim 35$, which is required to prevent pair production in the emission zone due to low-energy synchrotron photons as well as avoid a super Eddington jet, we deduce the size of the reconnection region $l' \sim 8.2 \times 10^{15}$ cm. The Doppler factor of the monster plasmoid $\delta_p \sim 85$ is computed by using the luminosity contrast between the minute-scale-flare and the envelope emission (detailed calculations can be found in "Methods"). The minute-scale-flare powered by monster plasmoids show an exponential rise and last for $t_p \sim (1+z)fl'/\delta_p c \sim 493 \times (f/0.1)(l'/8.2 \times 10^{15}$ cm$)/(\delta_p/85)$ s $\sim 8$ min, which is very close to the observed variability.

Considering the presence of a magnetic field of strength $\sim 7$ G in the magnetic reconnection region also fulfils the energy requirement of the jet at the time of the envelope emission and during the minute-scale-flare. With reference to the jet-in-jet model, the isotropic (equivalent) envelope and monster plasmoid luminosities are computed as $L_{env} \sim 7.0 \times 10^{48}$ erg s$^{-1}$ and $L_p \sim 2.6 \times 10^{49}$ erg s$^{-1}$, respectively. These values are in good agreement with the observed luminosities of envelope emission $L_{env,obs} \sim 6.7 \times 10^{48}$ erg s$^{-1}$ and minute-scale-flare emission $L_{msf,obs} \sim 2.1 \times 10^{49}$ erg s$^{-1}$ recorded during flare F2 (cf. detailed calculations in "Methods").

The substantial magnetic dissipation takes place where the external medium density begins to drop from its central plateau value outside of the BLR and where the reconnection time-scale becomes comparable to the expansion time-scale of the jet at a distance $R_{diss} \sim \Gamma_j^2 r_S/\epsilon$. Assuming black hole mass $\sim 4 \times 10^8 M_\odot$, $\epsilon \sim 0.1$ and $\Gamma_j \sim 35$, we obtain, $R_{diss} \sim 0.48$ pc, which also satisfies the $l' \leq R_{diss}\theta_j$ condition needed to keep the reconnection region inside the jet cross section. Here, $\theta_j$ is the jet opening angle, and it is related to jet Lorentz factor through the relation $\theta_j\Gamma_j \sim 0.2$ as suggested[38]. Thus, we conclude that the gamma-ray flare region lies outside of the BLR in 3C 279 in agreement with other recent studies[26,32,39–41]. Figure 2b shows a schematic diagram of the envisaged scenario: The minute-scale-flares superimposed on the more slowly varying envelope emission result from the magnetic reconnection of multiple twisted flux ropes in a turbulent, filamentary region where the magnetically dominated jet becomes unstable to the kink instability.

Mini-jets from a reconnection region, located where the jet collimation breaks down, produce optically thin gamma-ray emission mainly by the external Compton process and possibly by the photo-production of pions in interactions with UV photons from the BLR and IR photons from the dusty torus, shown in magenta colour in Fig. 2b. The curvature of the gamma-ray spectrum could reflect the decrease of the Klein-Nishina scattering cross section[42] with rising energy. Thus, the ultra-short variability seems to indicate processes associated with the transition of the magnetic-field-dominated jet to the kinetic-energy dominated jet.

## Methods

**Fermi-LAT data analysis**. The Fermi-LAT[29] is a pair-conversion gamma-ray telescope on board the Fermi spacecraft. It covers an energy range of 20–500 GeV with a 2.5 sr large field of view and can scan the entire sky in about 3 hours. The P8R3 Fermi-LAT gamma-ray data ( >100 MeV) of FSRQ 3C 279 has been analysed using the standard analysis procedure (Fermitools 1.2.1) provided by the Fermi-LAT collaboration and user-contributed Fermipy package[43]. A Region of Interest (ROI) with a circular radius of 15° around the 3C 279 was selected for analysis. A zenith angle cut of 90°, the GTMKTIME cut of DATA_QUAL > 0&&LAT_CONFIG==1 together with the evclass=128 and the evtype=3 was used. Spectral analysis on the resulting data set was carried out by including gll_iem_v07.fits and the isotropic diffuse model iso_P8R3_SOURCE_V2_v1.txt. The flux and spectrum of 3C 279 were determined by fitting a log parabola model, using a binned gtlike algorithm based on the NewMinuit optimiser[44,45]. The energy of highest energy photons was deduced using gtsrcprob tool on ULTRACLEAN event class (evclass=512).

The soft X-ray data from Neil Gehrels Swift-X-Ray Telescope (XRT) were reduced with standard methods, using the XRT data analysis software (XRTDAS7) distributed within HEASOFT version 6.19 and calibration databases. X-ray observations were carried out by Swift on 20 April 2018. The integral flux between 0.3 and 8 keV was found to be $\sim 4 \times 10^{-11}$ erg cm$^{-2}$s$^{-1}$. The X-ray flux is used to determine the minimum Lorentz factor which is needed to avoid pair-production in the emission zone due to low-energy synchrotron photons.

**Timing characteristics of gamma-ray flares from 3C 279**. During the high activity period between January and June 2018, 3C 279 had shown a few noteworthy flares including one of the brightest flares ever detected from any AGN by Fermi-LAT at gamma-ray energies. The flux history of the source above 100 MeV with one-day binning covering the period between MJD 58100 - MJD 58280, is shown in Fig. 1a using black circles, the only points having three-sigma detection are plotted. Three pronounced gamma-ray flares F1, F2, and F3 were identified and the their three-hour binned light curves are also plotted in Fig. 1a in different colours. The timing and spectral properties of all three flares were found to be very similar, see Table 1 and Table 2. The spectral indices of the flares are rather hard, and the similarity in spectral curvature strongly suggests that the flares might have originated from the same location in the jet, suffering exposure to the same external target photon fields for inverse-Compton scattering or photo-pion production. The hardest average spectrum was found during the third flare with the index $\alpha = 1.87 \pm 0.03$.

Out of these three flares, first two had shown characteristic peak-in-peak light curve, where rapidly varying fast flares were superimposed on slowly varying envelope emission of the order of a day scale. The fast flare in F1 lasted for a few hours between (MJD 58136.18 - MJD 58136.56) while in F2, a couple of ultra-bright fast flares FF1 and FF2 were recorded between (MJD 58227.76 - MJD 58228.40). Notably, the flare F3 did not show any noteworthy fast component. A magnified view of the light curve of flare F2 is presented in Fig. 1b and the average spectra of F1, F2 and F3 are shown Fig. 1c while the slowly varying envelope and average fast flare spectra of F2 are plotted in Fig. 1d. The error bars shown in the light curves represent 1 $\sigma$ uncertainty. The observed gap in the Fermi-LAT light curve is attributed to a technical issue encountered on 16 March 2018 by Fermi

**Table 2 Spectral properties of the flares observed from 3C 279 during 2018.**

| Flare Name | Epoch (MJD) | Flux × 10$^{-6}$ | $\alpha$ | $\beta$ | TS |
|---|---|---|---|---|---|
| F1 | 58133.0–58139.0 | 13.43 ± 0.25 | 2.07 ± 0.02 | 0.11 ± 0.01 | 12655 |
| F2 | 58222.0–58232.0 | 9.57 ± 0.11 | 1.97 ± 0.01 | 0.10 ± 0.01 | 45977 |
| F3 | 58268.0–58276.0 | 7.70 ± 0.20 | 1.87 ± 0.03 | 0.15 ± 0.02 | 6649 |
| Envelope [F2] | 58222–58227.76 & 58228.40–58232.00 | 8.53 ± 0.10 | 1.99 ± 0.01 | 0.10 ± 0.01 | 37149 |
| Fast flare [F2] | 58227.76–58228.40 | 24.97 ± 0.60 | 1.86 ± 0.03 | 0.14 ± 0.02 | 16339 |

Column (1) presents the flare name, Column (2) presents the observation duration, Column (3) presents the observed flux during each epoch in the units of Ph cm$^{-2}$ sec$^{-1}$, Column (4) presents the spectral index $\alpha$ at E$_b$ when fitted with logparabola model, here E$_b$ is the scale parameter, Column (5) presents the curvature parameter $\beta$ when fitted with logparabola model, Column (6) presents the test statistics (TS) of the detection.

spacecraft, see Fig. 1a. Fermi-LAT had not carried out any science observations between 16 March and 08 April 2018.

All three observed flares are quasi-symmetric in nature and a sum of two exponential functions are used to determine their rise and decay times. The fitted function is expressed as

$$F(t) = 2F_0 \left[ \exp\left(\frac{t_0 - t}{T_r}\right) + \exp\left(\frac{t - t_0}{T_d}\right) \right]^{-1} \qquad (1)$$

where $F_0$ is the flux at time $t_0$ representing the approximate flare amplitude, and $T_r$ and $T_d$ are the respective rise and decay times of the flare. To calculate the minimum flux doubling time between the time instants $t_1$ and $t_2$, each flare is scanned separately by using the following function,

$$F(t_2) = F(t_1) \cdot 2^{(t_2 - t_1)/\tau} \qquad (2)$$

where, $F(t_1)$ and $F(t_2)$ are the fluxes measured at $t_1$ and $t_2$ respectively, and $\tau$ represents the flux doubling timescale. Further, to study the timing properties of F1, F2, and F3, these flares are fitted with slowly varying envelope and fast flaring components using the above-mentioned function (Eq. 1). The details of the fit-parameters are provided in Table 1. The flare F1 is fitted with a slowly varying envelope and a fast flaring component yielding a reduced $\chi^2 \sim 1.8$ with the rise time of the envelope emission $\sim 1.62 \pm 0.06$ d. Similarly, when the flare F2 is fitted with a slowly varying envelope and two fast components, a reduced $\chi^2 \sim 1.5$ and rise time for envelope emission of $\sim 1.61 \pm 0.03$ d is obtained. Notably, flare F3 did not show any fast component. This flare is fitted with a slowly varying envelope which gave a reduced $\chi^2 \sim 1.7$ and the rise time of $\sim 1.47 \pm 0.08$ d. Moreover, the flare F2 was found to be very complex, see Fig. 1b, the overall slowly varying envelope can be decomposed as contributions from five small envelopes. Two very bright fast flares (FF1 and FF2) were found to be superimposed on the overall slowly varying envelope. We also noticed a low amplitude fast flare FF3 during envelope emission. The overall fit yielded a reduced of $\chi^2 \sim 1.1$. The rise time of the slowly varying component which acted as an envelope for fast flares had a rise time of $\sim 1.35 \pm 0.34$ d. The reduced $\chi^2$ values for the flare F2 with a combination of one envelope and two fast flares and with another combination of five envelopes and three fast flares were found to be comparable. The rise time of all fast flare events is found to be $\sim 0.1$ d.

Several high energy photons were detected by Fermi-LAT during F1, F2, and F3. All the photons of ULTRACLEAN class, which have a probability of association >99.99% with 3C 279, are plotted as violet bars (right $Y$-axis) in Fig. 1a.

**Minute-scale variability**. The 3C 279 was the first blazar to show strong and rapid variability at GeV energies from Compton Gamma Ray Observatory (CGRO)[25] and Fermi-LAT[11]. Between January and June 2018, the activity exhibited by 3C 279 was observed by Fermi-LAT and this provided an opportunity to resolve the gamma-ray light curve with a shorter timescale than the Fermi orbital period. A three-minute binned light curve, fitted with a constant flux, was used to investigate flux variability at sub-orbital timescales for flares F1, F2, and F3. The flux in most of the orbits was found to be consistent with constant flux. However, within F2, the source was highly inconsistent with a constant flux during a few orbits with p-values $\sim$ 0.002 ($\chi^2$-test), and 0.06 ($\chi^2$-test). During the orbit with p-value 0.002 ($\chi^2$-test), starting from MJD 58227.90 to MJD 58227.97 shown by a vertical solid grey line in Fig. 1b, we witnessed a strong few-minute scale variability (flux doubling timescale $\tau \sim$ 8 min) with a significance of 4.7 $\sigma$. However, when a ten-minute binned light curve was fitted with a constant flux in the same orbit (MJD 58227.90 to MJD 58227.97), we got a $p$-value $\sim 10^{-5}$ ($\chi^2$-test). Here, we report the detection of significant (6.6 $\sigma$) flux doubling timescale of $\sim 13$ min using ten-minute binned light curve. This corroborates further evidence of the sub-orbital variability of 3C 279. Moreover, we fitted a sum of two exponential functions presented in Eq. 1 to the three-minute binned light curve. The rise time was found to be $8.6 \pm 1.8$ min with the best fit, yielding a reduced $\chi^2$ of 0.95. The three-minute and 10-min binned light curves ( >100 MeV) of the aforementioned light curve, respectively plotted as circles and squares, are shown in Fig. 2a and the best-fit function to the three-minute light curve is represented by the dashed cyan-blue line in the same panel. Furthermore, the detection of a hard spectrum with index $\alpha = 1.74 \pm 0.08$, during the orbit in which minute-scale-flare was detected, strongly indicates particle acceleration through magnetic reconnection. Fermi-LAT also observed the one of the greatest flux from the source (above 100 MeV) during one single orbit $(3.4 \pm 0.2) \times 10^{-5}$ ph cm$^{-2}$ s$^{-1}$ of second fast flare of F2.

**Flare spectra**. A significant break or curvature in the gamma-ray spectrum is evident in powerful FSRQs during the flaring activity. The spectra of the slowly varying envelope phase and fast flare phase of F2 are extracted separately. The average spectrum of the slowly varying envelope was computed by adding data from MJD 58222.00 - MJD 58227.76 and MJD 58228.40 - MJD 58232.00 and plotted with blue in Fig. 1d. The three-hour binned light curve of this period is plotted in blue in Fig. 1b. The fast flare average spectrum was computed between MJD 58227.76 - MJD 58228.40 and plotted with maroon in Fig. 1d and the orbit binned light curve of this section is plotted in maroon in the Fig. 1b. The spectra for the slowly varying envelope and fast flare components are fitted with log-parabola (LP), a broken power law (BPL) and power law (PL) models. The LP

model is found significantly better than the PL and BPL models in describing the spectral shape and the data of F2. The measured spectra show significant curvature in both the fast flares and slowly varying envelope components. However, the measured fast flare spectrum is significantly harder than that of the slowly varying envelope component.

**Location and particle acceleration mechanism of gamma-ray emission**. During the flare F2, the spectral signature and temporal coincidence of the fast flares with the slowly varying envelope components help to constrain the location and nature of the particle acceleration zone. Differential Doppler boosting also has difficulties to explain the observed time profiles of the envelope and fast flare emission simultaneously with the spectral curvature. In the following, we evaluate different scenarios to explain the observed variability and spectra.

**Origin of fast flares and envelope emission inside the BLR**. If high-energy gamma-ray photons are emitted inside the BLR, they are expected to be absorbed by the UV photons radiated by H-$Ly_\alpha$ and continuum emission of the quasar due to a high optical depth for pair creation in photon-photon collisions[46]. The detection of high energy photons >13 GeV strongly suggests that at least part of the emission has originated either outside or at the edge of the BLR where the optical depth drops to lower values.

**Origin of fast flares inside BLR, envelope emission outside**. If the observed curvature in the spectra would result from the attenuation of gamma-rays by photon-photon pair production on H-$Ly_\alpha$ recombination lines within the BLR[46], or due to the Klein-Nishina effect and LP electron distribution[47], the observed similarity in the curvatures of the fast flare and envelope spectra renders this scenario unlikely since the optical depth of external photon scattering would be very different.

**Origin of fast flare and envelope emission outside the BLR**. This scenario can be further divided into two cases: non co-spatial and co-spatial.

*Non co-spatial*: detection of high-energy photons during the fast flare requires the emission region to be outside of the BLR. But for this to happen, the Lorentz factor of the jet must be around 100, a value which is theoretically demanding at parsec scales. However, even if we do consider such a high Lorentz factor in the flare region, the origin of envelope emission would be tens of a parsec away from the central machine. In this scenario, the observed spectra for both emission regions would probably be very different due to different levels of external photon fields, in contrast to the observational findings.

*Co-spatial*: this is the most viable scenario that has been discussed below in two-steps. Generally, the origin of blazar variability may be associated with the shocks. However, minute-scale variability from a re-confinement shock during fast flare F2 is difficult to reconcile with the shock-in-jet scenario. To better understand this, we estimate the cross-sectional radius of the re-confinement shock $R_{rcs} \sim 2.5 \times 10^{-2.5} \left(\frac{L_j}{10^{46} \text{ erg s}^{-1}}\right)^{-1} (d_{BLR}/0.1\text{pc})^{-1}$ pc[48]. Here, $d_{BLR}$ denotes the distance of the BLR from the central black hole. $L_j$ is the jet power and can be approximated as $\sim L_\gamma/\eta\Gamma_j^2$, where $L_\gamma$ is isotropic gamma-ray luminosity and $\eta$ the radiative efficiency of the jet, which is typically $\sim 0.1$ as suggested[49]. The gamma-ray luminosity $\sim 10^{49}$ erg s$^{-1}$, observed during the minute-scale-flare, strongly constrains the Lorentz factor of the jet. A lower limit of Lorentz factor is obtained by equating the radius of re-confinement shock with the size of the emission zone equivalent to eight minutes yielding the Lorentz factor of the jet $\Gamma_j \sim 335$, assuming $\Gamma_j \sim \delta_j$ and $\eta = 0.1$, where $\delta_j$ is the Doppler factor of the jet. This value is in contradiction with values found in kinematic studies of parsec-scale jets[50], and also with plausible magnetohydrodynamical (MHD) models of jets.

By contrast, the magnetic reconnection scenario provides a more appealing explanation for our two main findings: (a) the co-spatial origin of the fast flare and envelope emission as deduced by the observed similarity in their spectral shapes, and (b) the fast minute-scale variability. Magnetic reconnection naturally occurs in the turbulent plasma where the jet collimation weakens. The multi-scale nature of magnetic reconnection in a turbulent region provides a viable solution to the observed minute-scale variability, indicating structures much smaller than the jet diameter. At this location, the Poynting flux of the inner jet dissipates through topological rearrangement of the ordered field and the jet becomes kinetic energy dominated. The plasma providing the anomalous resistivity can be entrained from the surroundings or is created by pair production at the base of the jet. The magnetic field in the reconnection zone fragments and leads to the formation of a large number of smaller plasmoids. These plasmoids further grow into large, monster plasmoids which are responsible for the minute-scale variability[35]. The large-amplitude swings in polarization angle (PA) are expected during reconnection events. A significant change was observed in the PA from 3C 279 during the flare over the few days. Moreover, high-cadence observation of PA are needed to understand the role of PA change in such events during the fast flares. These high-cadence observations were unfortunately not available for the April 2018 flare.

The Lorentz factor of a monster plasmoid in the rest frame of the jet $\gamma_p$ can be estimated using the luminosity contrast between the minute-scale-flare to the envelope

emission, $\gamma_p = \delta_p/\Gamma_j \sim 2.2$. The Doppler factor of the monster plasmoid $\delta_p$ is a function of jet's Lorentz factor $\Gamma_j$, the monster plasmoid's Lorentz factor in the rest frame of the jet $\gamma_p$, and the angle of the monster plasmoid with respect to the jet's motion and the observer's angle of sight. Here, we adopt $\delta_p \sim 1.1\Gamma_j\gamma_p$ which is similar as used by Giannios et al.[35]. Considering $\Gamma_j \sim 35$ which is required to avoid pair production in the emission zone due to low-energy synchrotron photons and also to avoid a super Eddington jet, we compute the monster plasmoid's Doppler factor $\delta_p \sim 85$. The emission from the reconnection region as a whole can account for the observed envelope emission. The corresponding envelope timescales (rise time of envelope emission $t_{env} \sim 1.2 \times 10^5$ s) provide a characteristic size $l' = t_{env}\Gamma_j\epsilon c/(1+z) \sim 8.2 \times 10^{15}$ cm for the magnetic reconnection region in jet's co-moving frame. The size of monster plasmoids which may be responsible for minute scale variability may grow up to a sizeable fraction of 10% ($f = 0.1$) of the reconnection region. The rise time of minute-scale-flare produced due to the emission from the monster plasmoid can be expressed as $t_p \sim (1+z)fl'/\delta_p c \sim 493 \times (f/0.1)$ $(l'/8.2 \times 10^{15}$ cm$)/(\delta_p/85)$ s $\sim 8$ min, and these largest plasmoids have a recurrence interval of $\sim 2.5 (1+z)l'/\delta_p c$ which is consistent with the observed variability within fast flares during F2. The total envelope and monster plasmoid luminosities which are responsible for the envelope and minute-scale-flare emission, resp., in the jet-in-jet model can be expressed as

$$L_{env} = 2\Gamma_j^2\delta_p^2 l'^2 U_j'\epsilon c \text{ erg s}^{-1} \tag{3}$$

$$L_p = 4\pi f^2 l'^2 U_p'' c\delta_p^4 \text{ erg s}^{-1} \tag{4}$$

Here $\epsilon$ denotes the reconnection rate, $U_j'$ is the energy density at the dissipation zone in the co-moving frame of the jet, $U_p''$ is the energy density of the plasmoid in its co-moving frame. The magnetic dissipation takes place at distance $R_{diss} \sim 0.48$ pc, which also satisfies the $l' \leq R_{diss}\theta_j$ condition which is needed to keep the reconnection region inside the jet cross section. Here, $\theta_j$ is the jet opening angle, and it is related to jet Lorentz factor through the relation $\theta_j\Gamma_j \sim 0.2$, as suggested[38]. The isotropic envelope and monster plasmoid luminosities are found to be $L_{env} \sim 7.0 \times 10^{48}$ erg s$^{-1}$ and $L_p \sim 2.6 \times 10^{49}$ erg s$^{-1}$ using Eq. 3 and Eq. 4 respectively, considering $f = 0.1$, $\epsilon = 0.1$, $\delta_p \sim 85$, $U_p'' \sim U_j'$ and a magnetic field of strength $\mathbf{B} \sim 7$ G in reconnection region. Luminosities computed from the jet-in-jet model are in excellent agreement with the observed luminosities of the envelope emission $L_{env,obs} \sim 6.7 \times 10^{48}$ erg s$^{-1}$ and the minute-scale-flare emission $L_{msf,obs} \sim 2.1 \times 10^{49}$ erg s$^{-1}$ above 100 MeV.

**Reporting summary**. Further information on research design is available in the Nature Research Reporting Summary linked to this article.

## Data availability

The data used for this study are public and can be obtained through the Fermi-LAT data archives or data can also be obtained from the corresponding author upon request. Link for the data: https://fermi.gsfc.nasa.gov/cgi-bin/ssc/LAT/LATDataQuery.cgi.

## Code availability

Upon a reasonable request, the corresponding author will provide all code used for this study. The software used for data analysis are available on the links:
 Fermi-LAT: https://fermi.gsfc.nasa.gov/ssc/data/analysis/software/
 XRT: https://heasarc.gsfc.nasa.gov/docs/software/lheasoft/

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

## Acknowledgements

A.S. acknowledges support through BMBF grant 05A2017 - CTA/05A17WW1 (Verbundforschung). A.S. thanks Prof. A. R. Rao, Dr. V. C. Chitnis and Prof. A. K. Kembhavi for reading the paper. A.S. thanks Prof. G. C. Dewangan for the support provided during my visit to IUCAA. Open access funding provided by Projekt DEAL.

## Author contributions

A.S. conceived the idea and analysed the multiwavelength data, carried out timing and spectral analyses. A.S. and K.M. both contributed to the theoretical interpretation and jointly wrote the paper.

## Competing interests

The authors declare no competing interests.
