## [Peer Review File · Nature Communications]

Reviewers' comments:

Reviewer #1 (Remarks to the Author):

The manuscript deals with a major open question of high-energy astrophysics: the mechanisms behind the flares from blazar jets. It consists of two main parts. Firstly, the authors analyze observations of the quasar 3C 279 from the Fermi LAT telescope demonstrating convincingly that, over the first half of 2018, the source

exhibited a characteristic pattern of fast flares (lasting for minutes) superimposed with a slower-varying envelop of day-long emission. Then the authors proceed to explore the theoretical implications of their findings. They argue that the shock-in-jet model face

severe difficulties in accounting for the observations while a jet-in-jet model can account for the observed timescales and energetics of the flares for a reasonable choice of the model parameters. The authors conclude by sketching a coherent scenario for the inner blazar jet structure and emission. The manuscript is well-written, and, as far as I can tell, the analysis of the observational data and of the theoretical model is correct and accurate. The findings are of great importance for the field of research and with possible implications to other systems that contain relativistic flows. In my opinion, the manuscript is well worthy of publication to NatureCommunications after a few, minor issues are

addressed by the authors.

--line 38: please define, for completeness, the Schwarzschild radius r_s

--line 44: for simulations demonstrating the development of the kink instability in the jet, one can also refer to barniol-Duran et al. 2017, MNRAS, 469, 4957

--line 70: "fast-flares" should read "fast flares"

--line 97: The authors discuss that "formally a $\Gamma \sim 150$ may explain the minute-scale variability" in shock scenarios. The authors should provide a reference to support this claim. My impression is that the variability in shock scenarios cannot be arbitrarily short. It is limited by the variability introduced by the inner engine. The latter is closely related to the black hole size: $t_{\text{var}} > R_s/c$ (eg Spada et al. 2001, MNRAS, 325, 1559).

--line 125: the parameter f is not defined.

--line 132 and in several other locations: the subscripts and the units erg s^{-1} appear in math type (instead of Roman). Please correct throughout the text.

--line 134: the dissipation distance of 0.9pc is abruptly introduced. Where does this value come from? Is it related to the lifetime of the large plasmoid? In the supplemental material, the authors set $R_{\text{diss}}=2.9e18$ cm. Again the choice is not justified.

Reviewer #2 (Remarks to the Author):

3C 279 is one of the powerful gamma-ray source located at a redshift $z=0.536$. This source had gone through series of distinct flaring events and have been studied extensively over past decades. In the present work, authors study the minute scale variability from the source observed during 2018. They also identify the rapid flaring patterns ride over a slow flux variation and assert this as an evidence for the presence of magnetic re connection in blazar jets, further stretching it to address the conversion of Poynting flux to kinetic flux. However, such variations are very commonly seen in blazars and similar interpretation have been already been proposed. Further, its a weak interpretation which lack strong observational evidence and a theoretical basis. In my view, presenting a well known blazar feature and an uncertain interpretation do not qualify the present work to be published in Nature Communications. Hence, I do not recommend the manuscript for the publication. Below I list down few comments relevant to my decision:

Comments:

1.The minute scale variability in blazars are often witnessed in blazars and same interpretation is suggested earlier (Paliya et al 2015 ApJ 811,143). The only difference I see here is the source is different and study is performed at gamma ray energies instead of X-rays. The abstract in Paliya et al 2015 reads

"A pattern of extremely fast variability events superposed on slowly varying flares is found in most of the NuSTAR observations. We suggest that these peculiar variability patterns may be explained by magnetic energy dissipation and re connection in a fast-moving compact emission region within the jet."

The presented work is not very different from the Paliya et al. 2015.

2. In the work, the authors have claimed that the location of emission region should be outside the BLR region to explain the observational results. However, same results have been observed in several previous works for the source 3C 279 e.g, "Shah et al 2019, MNRAS, 484, 3168–3179", "Vitorini et al 2017, ApJ, 843, L23", "Dermer et al. 2014, ApJ, 782, 82" etc. Again, there is nothing new in this result. Further, the sub-orbital minute time scale variability at gamma-ray energy reported in the manuscript is not new for 3C 279. "Ackermann et al. 2016, ApJ, 824, L20" has reported a significant flux variability at sub-orbital time-scales with flux doubling times of less than 10 minutes, and plausibly ~5 minutes or shorter.

Other Issues:

1. The abstract is a bit blatant: The authors have tried to give twist in the manuscript by starting the abstract with the recent discovery of black hole image observed with the Event Horizon Telescope. However, I don't find the relevance of the observational results in the manuscript with the black hole image or with the Event Horizon Telescope observation in the rest of the manuscript.

2. The manuscript lacks proper citation of previous works on 3C 279:

There is no introduction to the previous flares of 3C 279 especially the 2015 June flare that was the brightest gamma-ray flare above 100 MeV of 3C 279 ever observed to date (for daily binned light curve). Also, during 2015 a significant flux variability at sub-orbital time-scales (~5 min) was observed by Fermi-LAT for the first time for 3C 279.

3. In the manuscript the authors have constrained the location of emission region outside the BLR region by using the argument of detection of high-energy photons above 13 GeV to avoid pair absorption. Is this really a good tool to constrain emission region.

Reason of my concern: To explain extremely short variability timescale of 2015 June flare, Ackermann et al 2016 located the emission region within BLR region even though the highest-energy photon of 56 GeV was observed from 3C 279 during 2015, June flare.

4. In this manuscript a 10 degree "Region of Interest" (ROI) around the 3C 279 is used for the analysis. However due to presence of a variable source 3C 273 at 10.336 degrees, a 15 degrees of ROI is usually suggested for the 3C 279.

Response to reviewer's comments: "Gamma-ray flares from relativistic magnetic reconnection in the jet of the quasar 3C 279"

A. Shukla^{1,*}, K. Mannheim¹

¹*Institut für Theoretische Physik und Astrophysik, Universität Würzburg, Emil-Fischer-Str. 31, 97074 Würzburg, Germany*

We thank both referees for their careful reading of the manuscript and their many constructive suggestions. We have answered all the issues raised by the referees. Reviewer's comments are given in purple color, and the author's reply is provided in black color in the report. All the changes are marked in the boldface in the manuscript and also provided in this report with blue text.

*amit008@gmail.com

1 Reviewers' comments:

Reviewer 1 (Remarks to the Author):

The manuscript deals with a major open question of high-energy astrophysics: the mechanisms behind the flares from blazar jets. It consists of two main parts. Firstly, the authors analyze observations of the quasar 3C 279 from the Fermi-LAT telescope demonstrating convincingly that, over the first half of 2018, the source exhibited a characteristic pattern of fast flares (lasting for minutes) superimposed with a slower-varying envelope of day-long emission. Then the authors proceed to explore the theoretical implications of their findings. They argue that the shock-in-jet model faces severe difficulties in accounting for the observations while a jet-in-jet model can account for the observed timescales and energetics of the flares for a reasonable choice of the model parameters. The authors conclude by sketching a coherent scenario for the inner blazar jet structure and emission. The manuscript is well-written, and, as far as I can tell, the analysis of the observational data and of the theoretical model is correct and accurate. The findings are of great importance for the field of research and with possible implications to other systems that contain relativistic flows. In my opinion, the manuscript is well worthy of publication to Nature Communications after a few, minor issues are addressed by the authors.

–line 38: please define, for completeness, the Schwarzschild radius r_s

Author's reply: Now Schwarzschild radius r_s is defined inline number 38.

–line 44: for simulations demonstrating the development of the kink instability in the jet, one can also refer to Barniol-Duran et al. 2017, MNRAS, 469, 4957

Author's reply: Barniol-Duran et al. 2017, MNRAS, 469, 4957 is referred in the manuscript inline number 46.

–line 70: "fast-flares" should read "fast flares"

Author's reply: "fast-flares" has been changed to "fast flares" throughout the manuscript as recommended by the referee.

–line 97: The authors discuss that "formally a Gamma 150 may explain the minute-scale Variability" in shock scenarios. The authors should provide a reference to support this claim. My impression is that the variability in shock scenarios cannot be arbitrarily short. It is limited by the variability introduced by the inner engine. The latter is closely related to the black hole size: $t_{var} R_s/c$ (eg Spada et al. 2001, MNRAS, 325, 1559).

Author's reply: We agree, and we have already omitted this sentence in the revised version

to avoid misunderstandings. Causality indeed limits the variability time scale associated with shocks traveling along the jet to be larger than the light travel time across the event horizon of the central supermassive black hole independent of the bulk Lorentz factor (Spada et al. 2001, MNRAS, 325, 1559). A few lines are included from line number 103-106 in main the text, and the same text is copied below in blue.

”Moreover, the observed peak-in-peak light curve can not be explained within the shock-in-jet scenario as the variability time scale would have to be larger than the light crossing time across the jet radius which itself exceeds the Schwarzschild radius³¹.”

–line 125: the parameter f is not defined.

Author’s reply: The parameter f is now defined in the main text inline number 119-121. Same text is copied below in blue.

This is the reconnection region in which some plasmoids can grow to a sizeable fraction of 10% ($f=0.1$) of the reconnection region through mergers before leaving the region and becoming monster plasmoids³³.

–line 132 and in several other locations: the subscripts and the units ergs^{-1} appear in math type (instead of Roman). Please correct throughout the text.

Author’s reply: The problem with the text is fixed, and now erg s^{-1} appear in Roman.

–line 134: the dissipation distance of 0.9pc is abruptly introduced. Where does this value come from? Is it related to the lifetime of the large plasmoid? In the supplemental material, the authors set $R_{diss}=2.9e18$ cm. Again the choice is not justified.

Author's reply: In the earliest version of the manuscript, we used the dissipation distance from the central engine estimated from $d \sim 2c\Gamma^2\Delta t/(1+z)$ (Abdo et al 2011, ApJL, 733, L26) which is related to internal shocks. This value can be considered as an upper limit on the distance from the central black hole. According to Giannios, D. MNRAS 431, 355-36336 (2013), the emission region due to relativistic reconnection should be found within a distance of ~ 0.1 pc to 0.9 pc from the central black hole. Here, we have determined the magnetic dissipation distance by requiring that the time scale for reconnection becomes comparable to the expansion time-scale of the jet. Denoting the jet Lorentz factor with Γ_j , the reconnection speed parameter with epsilon, and the Schwarzschild radius with r_s , we obtain $R_{diss} \sim \Gamma_j^2 r_s / \epsilon$. This is the relevant distance scale for the jet-in-jet model. Now, assuming a black hole mass of $\sim 4 \times 10^8 M_\odot$, $\Gamma_j=34$, and $\epsilon = 0.1$, we obtained the value $R_{diss} \sim 0.46$ pc. This value also satisfies the $l' \leq R_{diss}\theta_j$ condition required to keep the reconnection region smaller than the jet cross section (l' denotes the characteristic size of the magnetic reconnection region and θ_j is jet opening angle). The minimum R_{diss} must be 0.44 pc away from the central engine. A paragraph on dissipation distance is added in the manuscript between line number 142-149. The same text is copied below in blue for the reference.

The substantial magnetic dissipation takes place where the external medium density begins to drop from its central plateau value outside of the BLR and where reconnection time-scale becomes

comparable to the expansion time-scale of the jet at a distance $R_{diss} \sim \Gamma_j^2 r_s / \epsilon$. Assuming black hole mass $\sim 4 \times 10^8 M_\odot$, $\epsilon \sim 0.1$ and $\Gamma_j \sim 34$, we obtained, $R_{diss} \sim 0.46$ pc, which also satisfies the $l' \leq R_{diss} \theta_j$ condition which is needed to keep the reconnection region inside the jet cross section, here, θ_j is the jet opening angle, and it is related to jet Lorentz factor through the relation $\theta_j \Gamma_j \sim 0.2$ as suggested³⁵. Thus we conclude that the gamma-ray flare region lies outside of the BLR in 3C 279 in agreement with other recent studies^{27,30,36–38}.

Reviewer 2 (Remarks to the Author):

3C 279 is one of the powerful gamma-ray source located at a redshift $z=0.536$. This source had gone through series of distinct flaring events and have been studied extensively over past decades. In the present work, authors study the minute scale variability from the source observed during 2018. They also identify the rapid flaring patterns ride over a slow flux variation and assert this as an evidence for the presence of magnetic re connection in blazar jets, further stretching it to address the conversion of Poynting flux to kinetic flux. However, such variations are very commonly seen in blazars and similar interpretation have been already been proposed. Further, its a weak interpretation which lack strong observational evidence and a theoretical basis. In my view, presenting a well known blazar feature and an uncertain interpretation do not qualify the present work to be published in Nature Communications. Hence, I do not recommend the manuscript for the publication.

Author's reply: We think that the 2nd referee's assessment is based on a biased weighing of the evidence we bring forward. It is true that fast gamma-ray variability was found in various

blazars; however, the characteristic peak-in-peak variability pattern is detected for the first time on time scales of minutes. Moreover, only because of the high flux density of 3C 279 we were able to discover the conjunction of minute-scale variability, peak-in-peak lightcurve, and absence of pair attenuation as the unique signature of magnetic reconnection outside of the BLR. Such rare events are of utmost diagnostic value and provide a unique opportunity to advance our scientific insight into the chain of processes responsible for the dissipation of the energy carried by jets. The finding ventures new insights in line with detailed theoretical predictions about magnetic reconnection and the kink instability in jets, and reaches far beyond earlier findings reported by the Fermi Collaboration (Ackermann et al. 2016, ApJ, 824, L20).

Below I list down few comments relevant to my decision:

Comments: 1. The minute scale variability in blazars are often witnessed in blazars and same interpretation is suggested earlier (Paliya et al 2015 ApJ 811,143). The only difference I see here is the source is different and study is performed at gamma ray energies instead of X-rays. The abstract in Paliya et al 2015 reads "A pattern of extremely fast variability events superposed on slowly varying flares is found in most of the NuSTAR observations. We suggest that these peculiar variability patterns may be explained by magnetic energy dissipation and re connection in a fast-moving compact emission region within the jet." The presented work is not very different from the Paliya et al. 2015.

Author's reply: We believe that the claimed ~ 14 min X-ray variability from NuStar data by the Paliya et al. 2015 from the Mrk 421 on the 11 April 2013 is an overestimation. The observations

reported by Paliya et al., where ~ 14 min variability is claimed, is based on binning the NuStar data in a bin size of 5 minutes and examining the fastest bin-to-bin variation. We have analysed the NuStar data of OBS id "60002023025" where ~ 14 min X-ray variability is claimed using `nupipeline` version 0.4.6 distributed with HEASOFT* v6.25, and calibration files 20181030, to create clean event files. For comparison, we have provided a light curve obtained from Paliya et al. 2015 in the panel "a" and light curve obtained from our analysis (FPMA and FPMB telescope combined) in the panel "b" of Fig 1. The light curve provided in the panel "c" is a standard product of the NuStar FPMB telescope. Moreover, the panel "d" shows the expanded view of the light curve. There was a ~ 40 min data gap between 88.125 - 90.625 ks. We think that authors did not notice this data gap by mistake and considered two adjacent bin time difference to be 5 min, which had resulted in an apparent sub-hour scale (~ 14 min) variability in the Mrk 421. However, the conclusively observed variability must be the order of an hour scale or more during the decay phase of the flare between 90.625 - 92.20 ks on 11 April 2013.

However, the tentative findings of Paliya et al. 2015 apply to the X-ray emission of a BL Lacertae object. The observed hard X-ray variability timescale from Mrk 421 as referred in Paliya et al. 2018 are found to be larger than light crossing timescale across the black holes event horizon. Which is about 15 min considering black hole mass of Mrk 421 is $1.9 \times 10^8 M_{\odot}$ (Barth, A. J., Ho, L. C., Sargent, W. L. W. 2003, ApJ, 583, 134). The observed hard X-ray flares can be explained either by conventional shock-in-jet blazar models where a shock crosses the emission region in the jet or through multiple injections of particles in the emission zone. Therefore, the observed hard

*<https://heasarc.gsfc.nasa.gov/docs/software/heasoft/>

X-ray variability by NuStar from Mrk 421 does not warrant the magnetic reconnection scenario. Moreover, the luminosity contrast between envelope and flares from NuStar is ~ 1 , suggesting similar Doppler factor for envelope and flares. A few times higher Doppler factor for flare is needed to explain sub-hour scale variability in the jet-in-jet model. Therefore, the jet-in-jet model does not explain the observed behavior of NuStar observations, even though the observed light curve appears to be similar to what is expected from the jet-in-jet model scenario. On the other hand, the superposition of emission due to shock accelerated particle from multiple blobs present in the jet can easily explain the observed pattern in the X-ray light curve of Mrk 421.

By contrast, we have found that much more complex and unique gamma-ray variability pattern in the light curve of a flat-spectrum radio quasar. The peak-in-peak light curve displays the ratio of half-widths between the short-term flares and the longer duration envelope emission as well as the isotropic luminosities theoretically predicted by Giannios 2013 for magnetic dissipation in blazar jets. Pair absorption constraints from the BLR let us conclude that the dissipation occurs outside of the BLR, whereas the localization of the putative X-ray dissipation region claimed by Paliya et al. 2015 is comparatively unconstrained.

Figure 1: A light curve obtained from Paliya et al. 2015 shown in the panel "a" for OBS id "60002023025" where ~ 14 min variability is claimed by Paliya et al. on 11 April 2013. The light curve obtained by our analysis (FPMA and FPMB telescope combined) is presented in the panel "b" for the same OBS id "60002023025". The light curve provided in the panel "c" is a standard product of the NuStar FPMB telescope for same OBS id "60002023025". And finally, panel "d" shows the expanded view of the light curve between 82.5-100 Ksec for the same OBS id "60002023025". A clear data gap of ~ 40 min is present at 88.125 to 90.625 ks in the light curve, where ~ 14 minute variability is claimed by Paliya et al. We think that authors did not notice this data gap by mistake and considered two adjacent bin time difference to be 5 min, which had resulted in an apparent sub-hour scale (~ 14 min) variability in the Mrk 421.

2. In the work, the authors have claimed that the location of emission region should be outside the BLR region to explain the observational results. However, same results have been observed in several previous works for the source 3C 279 e.g, "Shah et al 2019, MNRAS, 484, 31683179", "Vitorini et al 2017, ApJ, 843, L23", "Dermer et al. 2014, ApJ, 782, 82" etc. Again, there is nothing new in this result. Further, the sub-orbital minute time scale variability at gamma-ray energy reported in the manuscript is not new for 3C 279. "Ackermann et al. 2016, ApJ, 824, L20" has reported a significant flux variability at sub-orbital time-scales with flux doubling times of less than 10 minutes, and plausibly 5 minutes or shorter.

Author's reply: Ackermann et al. (2016), argue that the dissipation region is located at 200 gravitational radii without rigorously applying constraints from the pair opacity of the BLR, owing to the fact that photon statistics are low. However, it is known from flares observed with MAGIC (MAGIC Collaboration, Science, Volume 320, Issue 5884, pp. 1752- (2008)) and HESS (HESS Collaboration, 2019arXiv190604996H) that the gamma-ray spectra extend up to a few 100 GeV energies, and the flares described in our paper also exhibit multiple photons well above 10 GeV. The peak-in-peak pattern in the light curve together with the absence of pair attenuation shows for the first time convincingly that magnetic dissipation occurs outside of the BLR. Which have not been addressed by any previous work such as Ackermann et al. 2016, ApJ, 824, L20, "Shah et al 2019, MNRAS, 484, 31683179", "Vitorini et al 2017, ApJ, 843, L23", and "Dermer et al. 2014, ApJ, 782, 82".

Other Issues: 1. The abstract is a bit blatant: The authors have tried to give twist in the manuscript by starting the abstract with the recent discovery of black hole image observed with the Event Horizon Telescope. However, I don't find the relevance of the observational results in the manuscript with the black hole image or with the Event Horizon Telescope observation in the rest of the manuscript.

Author's reply: We have rephrased the abstract to avoid this misconception. The relevance of the EHT observations is that they claim evidence for a magnetically dominated jet in M87, corroborating the need for a mechanism by which the magnetic power is converted to kinetic power in the jet.

2. The manuscript lacks proper citation of previous works on 3C 279: There is no introduction to the previous flares of 3C 279 especially the 2015 June flare that was the brightest gamma-ray flare above 100 MeV of 3C 279 ever observed to date (for daily binned light curve). Also, during 2015 a significant flux variability at sub-orbital time-scales (~ 5 min) was observed by Fermi-LAT for the first time for 3C 279.

Author's reply: We have included a detailed text mentioning the earlier report of fast variability from the source 3C 279 during 2015 as suggested by the referee in the main text at line number 60-63. The added text is given below in blue.

During the Fermi-LAT era, several strong outbursts²⁵⁻²⁷, including the one of June 2015 were observed. This outburst showed flux doubling in just a few minutes reaching an isotropic luminosity

of 10^{49} erg s^{-1} in its maximum¹¹.

A few more citation on previous works and variability studies of the 3C 279 are added to the manuscript. The added citations are given below:

1. Hartman, R. C., Bertsch, D. L., Fichtel, C. E., et al. 1992, ApJL, 385, L1
2. Kniffen, D. A., Bertsch, D. L., Fichtel, C. E., et al. 1993, ApJ, 411, 133
3. Hayashida, M., Madejski, G. M., Nalewajko, K., et al. 2012, ApJ, 754, 114
4. Hayashida, M., Nalewajko, K., Madejski, G. M., et al. 2015, ApJ, 807, 79
5. Wehrle, A. E., Pian, E., Urry, C. M., et al. 1998, ApJ, 497, 178
6. Paliya, V. S. 2015, ApJL, 808, L48

3. In the manuscript the authors have constrained the location of emission region outside the BLR region by using the argument of detection of high-energy photons above 13 GeV to avoid pair absorption. Is this really a good tool to constrain emission region. Reason of my concern: To explain extremely short variability timescale of 2015 June flare, Ackermann et al 2016 located the emission region within BLR region even though the highest-energy photon of 56 GeV was observed from 3C 279 during 2015, June flare.

Author's reply: The Ackermann et al.(2016) considered the location of the gamma-ray emission region at 200 gravitational radii without rigorously applying constraints from the pair

opacity of the BLR because photon statistics are low. However, we have detected a good number of photons above 13 GeV (see Fig.1 of the manuscript) and even a few photons above 90 GeV during these flares, which allow us to constrain the location of the gamma-ray emission. Moreover, detection of very high energy gamma-ray emission from 3C 279 by HESS telescope during the active period in 2018 strongly advocate that gamma-ray emission should have been originated outside of the BLR. Recently H. E. S. S. Collaboration et al. 2019 considered shell and ring geometry of BLR for 3C 279 and showed that gamma-ray emission region must be located outside of the BLR. Most authors agree, and simple estimates show that the pair-creation opacity of the BLR in 3C 279 excludes the possibility that the highest energy gamma-rays observed with Fermi-LAT are produced within the BLR, a few references which discussed this issue are given below for 3C 279 and other FSRQs and cited in the manuscript.

1. H. E. S. S. Collaboration, 2019arXiv190604996H (accepted in A&A for publications)
2. Michael Zacharias, 2019, PoS(HEASA2018)033
3. Böttcher, Markus; Els, Paul 2016, ApJ, 821, 102B
4. Paliya V. S., 2015, ApJL, 808, L48
5. Liu, H. T., & Bai, J. M. 2006, ApJ, 653, 1089
6. Shah et al 2019, MNRAS, 484, 31683179
7. Dermer et al. 2014, ApJ, 782, 82
8. Meyer M., Scargle J. D., Blandford R. D., 2019, ApJ, 877, 39

9. MAGIC Collaboration, Science, Volume 320, Issue 5884, pp. 1752- (2008)
10. Pacciani, L., Tavecchio, F., Donnarumma, I., et al. 2014, ApJ, 790, 45
11. Aleksic, J., Antonelli, L. A., Antoranz, P., et al. 2011, ApJL, 730, L8
12. Shukla et al. 2018, ApJL, 854:L26
13. A. A. Abdo 2011, ApJ, 733:L26

4. In this manuscript a 10 degree "Region of Interest" (ROI) around the 3C 279 is used for the analysis. However due to presence of a variable source 3C 273 at 10.336 degrees, a 15 degrees of ROI is usually suggested for the 3C 279.

Author's reply: The source 3C 273 was in a quiescent state during MJD 58100 to MJD 58300 and did not affect the flux and spectrum estimation of 3C 279. A long term Fermi-LAT light curve of 3C 273 with one day bin is shown in Fig 2. A zoom light curve of 3C 273 from MJD 58200 onward is also shown in Fig 3, which indicate that the source is hardly detected with one day bin during the period of our interest (MJD 58100 - MJD 58300). The presented light curves can be found at the Fermi-LAT website. We believe that considering 10° "Region of Interest" (ROI) around the 3C 279 is sufficient. We would wish to retain all the results obtained using the 10° ROI in the manuscript.

However, for comparison, we have analyzed Fermi-LAT data of 3C 279 taking 15° ROI with the unbinned and binned likelihood method using newly introduced `Fermitool` package

Figure 2: Long term LAT (~ 11 years) light curve of 3C 273 is shown above. The source 3C 273 was in a quiescent state during MJD 58100 to MJD 58300, which is the period of interest for our work and had no effect on the results. This light curve can be found at: https://fermi.gsfc.nasa.gov/FTP/glast/data/lat/catalogs/asp/current/lightcurves/3C273_86400.png

(Conda) along with the most recent instrument response functions P8R3_SOURCE_V2. We have also included 3C 273 in the analysis.

No significant changes are found in the observed flux, light curves, and spectra obtained using 15° ROI in comparison with analysis performed using 10° ROI. A few residual maps which are produced through binned analysis are shown in Figures 7, 8, and 9. No differences have been found in the observed flux and spectra with the unbinned and binned analysis.

Figure 3: Zoomed light curve of 3C 273 between MJD 58200 - MJD 58700 is shown in the above plot. The source 3C 273 was barely detected during one-day exposure MJD 58200 - MJD 58300. This light curve can be found at : https://fermi.gsfc.nasa.gov/FTP/glast/data/lat/catalogs/asp/current/lightcurves/3C273_86400_1yr.png

Figure 4: Three hours binned light curves generated using 10° ROI in blue (Data: P8R2, IRF: P8R2_SOURCE_V6) and 15° ROI in red (Data: P8R3, IRF: P8R3_SOURCE_V2) using an unbinned gtlike algorithm are compared in the above plot. No significant changes have been observed between two light curves.

Figure 5: Three minute binned light curves generated using 10° ROI in blue (Data: P8R2, IRF: P8R2_SOURCE_V6) and 15° ROI in red (Data: P8R3, IRF: P8R3_SOURCE_V2) using an unbinned gtlike algorithm are compared in the above plot. No significant changes have been observed between two light curves.

Figure 6: **Left:** Spectral energy distribution of flare "F2" generated using 10° ROI is presented in blue (Data: P8R2, IRF: P8R2_SOURCE_V6) and 15° ROI in black (Data: P8R3, IRF: P8R3_SOURCE_V2) using an unbinned glike algorithm; **Right:** Spectral energy distribution of fast flare generated using 10° ROI is presented in red (Data: P8R2, IRF: P8R2_SOURCE_V6) and 15° ROI in black (Data: P8R3, IRF: P8R3_SOURCE_V2) using an unbinned glike algorithm. No significant changes have been observed among the SEDs obtained through 10° and 15° ROI.

Figure 7: The residual map of the ROI when 3C 279 and 3C 273 are not included in the model and contribution of both the sources are present in the map.

Figure 8: The residual map of the ROI when 3C 273 is not included in the model is presented in this plot. The contribution from 3C 273 is visible at the upper right corner of the residual map.

Figure 9: The residual map of the ROI is shown when both sources 3C 279 and 3C 273 are included in the model.

No significant contribution from any of the source is present in the residual map.

2 Additional changes

A Lorentz factor of the jet $\Gamma_j \sim 34$ have been adopted now to model the jet-in-jet scenario, which is required to avoid a super Eddington jet (earlier we had not taken this effect in the account) and pair production in the emission zone due to low-energy synchrotron photons. The Doppler factor of the monster plasmoid $\delta_p \sim 85$ is re-computed by using the luminosity contrast between the minute-scale-flare and the envelope emission (detailed calculations can be found in *Methods*).

The minute-scale variability from a re-confinement shock during fast-flare "F2" is difficult to reconcile with the shock-in-jet scenario. A lower limit of Lorentz factor is obtained by equating the radius of re-confinement shock with the size of the emission zone equivalent to five minutes yielding the Lorentz factor of the jet $\Gamma_j \sim 390$, assuming $\Gamma_j \sim \delta_j$ and $\eta = 0.1$ (**earlier $\eta = 1$ value was considered however $\eta = 0.1$ is more appropriate (Nemmen, R. S. et al. 2012, Science, 338, 1445)**, where δ_j is the Doppler factor of the jet. This value is in contradiction with values found in kinematic studies of parsec-scale jets (Jorstad, S. G. et al. AJ, 130, 1418-1465 (2005)), and also with plausible MHD models of jets.

We have also corrected a citation; earlier we had cited "Blandford, R. D., Payne, D. G. Hydromagnetic flows from accretion discs and the production of radio jets. MNRAS, 199, 883-903 (1982)" in place of "Blandford R. D., Znajek R. L., "Electromagnetic extraction of energy from Kerr black holes", MNRAS, 179, 433-456 (1977)".

The changes which are incorporated in the main text and supplementary text of the manuscript

are provided in the next two sections in blue.

Changes in the main text

1. Line number:119-121: Denoting the variability time-scale as $\Delta t = 60\Delta t_{\min}$ s, and assuming incoherent emission from a region of size r moving with bulk Lorentz factor $\Gamma_j = 34\Gamma_{34}$ at a cosmological redshift z , causality requires that in units of the Schwarzschild radius $r_S = 2GM/c^2$ of a black hole with mass M , the region must be smaller than $r/r_S = 2.0m_8^{-1}\Gamma_{34}\Delta t_{\min}(1+z)^{-1}$. In blazars, the black hole masses are typically very large such that $m_8 = M/10^8M_\odot > 1$ where M_\odot denotes one solar mass, i.e. the gamma-ray emitting regions must be smaller than the black hole horizon.

2. Line number:124-134: The characteristic size of the magnetic reconnection region in jet's co-moving frame can be expressed as $l' = t_{env}\Gamma_j\epsilon c/(1+z)$. Here ϵ parametrizes the reconnection speed; with $\epsilon \sim 0.1$ been a typical observationally inferred value. Further, by adopting a Lorentz factor of the jet $\Gamma_j \sim 34$, which is required to avoid pair production in the emission zone due to low-energy synchrotron photons and also to avoid a super Eddington jet, and $\epsilon \sim 0.1$, we deduce the size of the reconnection region $l' \sim 7.96 \times 10^{15}$ cm. The Doppler factor of the monster plasmoid $\delta_p \sim 85$ is computed by using the luminosity contrast between the minute-scale-flare and the envelope emission (detailed calculations can be found in *Methods*). The minute-scale-flare powered by monster plasmoids show an exponential rise and last for $t_p \sim (1+z)fl'/\delta_p c \sim 480 \times (f/0.1)(l'/7.96 \times 10^{15} \text{ cm})/(\delta_p/85)$ s ~ 8 min which is very close to observed variability.

3. Line number:135-139:Considering the presence of a magnetic field of strength ~ 7 G in the magnetic reconnection region also fulfils the energy requirement of the jet at the time of the envelope emission and during the minute-scale-flare. With reference to the jet-in-jet model, the isotropic envelope and monster plasmoid luminosities are computed as $L_{env} \sim 6.4 \times 10^{48} \text{ erg s}^{-1}$ and $L_p \sim 2.5 \times 10^{49} \text{ erg s}^{-1}$ respectively.
4. Line number:142-149: The substantial magnetic dissipation takes place where the external medium density begins to drop from its central plateau value outside of the BLR and where reconnection time-scale becomes comparable to the expansion time-scale of the jet at a distance $R_{diss} \sim \Gamma_j^2 r_s / \epsilon$. Assuming black hole mass $\sim 4 \times 10^8 M_\odot$, $\epsilon \sim 0.1$ and $\Gamma_j \sim 34$, we obtained, $R_{diss} \sim 0.46 \text{ pc}$, which also satisfies the $l' \leq R_{diss} \theta_j$ condition which is needed to keep the reconnection region inside the jet cross section, here, θ_j is the jet opening angle, and it is related to jet Lorentz factor through the relation $\theta_j \Gamma_j \sim 0.2$ as suggested³⁵. Thus we conclude that the gamma-ray flare region lies outside of the BLR in 3C 279 in agreement with other recent studies^{27,30,36-38}.

Changes in the Supplementary text

1. Line number:31:the only points having three-sigma detection are plotted.
2. Line number:148-153: A lower limit of Lorentz factor is obtained by equating the radius of re-confinement shock with the size of the emission zone equivalent to five minutes yielding the Lorentz factor of the jet $\Gamma_j \sim 390$, assuming $\Gamma_j \sim \delta_j$ and $\eta = 0.1$, where δ_j is the Doppler factor of the jet. This value is in contradiction with values found in kinematic studies of parsec-scale jets[?], and also with plausible magnetohydrodynamics (MHD) models of jets.
3. Line number:170-177: Here, we adopt $\delta_p \sim 1.0\Gamma_j\gamma_p$ which is the similar as used by Giannios et al.³³. Considering $\Gamma_j \sim 34$ which is required to avoid pair production in the emission zone due to low-energy synchrotron photons and also to avoid a super Eddington jet, we compute the monster plasmoid's Doppler factor $\delta_p \sim 85$. The emission from the reconnection region as a whole can account for the observed envelope emission. The corresponding envelope timescales (rise time of envelope emission $t_{env} \sim 1.2 \times 10^5$ s) provide a characteristic size $l' = t_{env}\Gamma_j\epsilon c/(1+z) \sim 7.96 \times 10^{15}$ cm for the magnetic reconnection region in jet's co-moving frame.
4. Line number:179-181:The rise time of minute-scale-flare produced due to the emission from the monster plasmoid can be expressed as $t_p \sim (1+z)fl'/\delta_p c \sim 480 \times (f/0.1)(l'/7.96 \times 10^{15} \text{ cm})/(\delta_p/85) \text{ s} \sim 8 \text{ min}$,
5. Line number:187-194: The magnetic dissipation takes place at distance $R_{diss} \sim 0.46 \text{ pc}$,

which also satisfies the $l' \leq R_{diss}\theta_j$ condition which is needed to keep the reconnection region inside the jet cross section. Here, θ_j is the jet opening angle, and it is related to jet Lorentz factor through the relation $\theta_j\Gamma_j \sim 0.2$, as suggested³⁵. The isotropic envelope and monster plasmoid luminosities are found to be $L_{env} \sim 6.4 \times 10^{48}$ erg s⁻¹ and $L_p \sim 2.5 \times 10^{49}$ erg s⁻¹ using Eq.3 and Eq.4 respectively, considering $f = 0.1$, $\epsilon = 0.1$, $\delta_p \sim 85$, $U_p'' \sim U_j'$ and a magnetic field of strength $B \sim 7$ G in reconnection region.

Bibliography

11. Ackermann, M. et al. Minute-timescale > 100 MeV γ -Ray variability during the giant outburst of quasar 3C 279 observed by Fermi-LAT in 2015 June. *Astrophys. J. Lett.* **824**, L20 (2016).
25. Hayashida, M. et al. The Structure and Emission Model of the Relativistic Jet in the Quasar 3C 279 Inferred from Radio to High-energy γ -Ray Observations in 2008-2010. *Astrophys. J.* **754**, 114 (2012).
26. Hayashida, M. et al. Rapid Variability of Blazar 3C 279 during Flaring States in 2013-2014 with Joint Fermi-LAT, NuSTAR, Swift, and Ground-Based Multiwavelength Observations. *Astrophys. J.* **807**, 79 (2015).
27. Paliya V. S. Fermi-Large Area Telescope Observations of the Exceptional Gamma-Ray Flare from 3C 279 in 2015 June. *Astrophys. J. Lett.* **808**, L48 (2015).
30. Liu, H. T. & Bai, J. M. Absorption of 10-200 GeV gamma rays by radiation from broad-line regions in blazars. *Astrophys. J.* **653**, 1089-1097 (2006).

31. Spada M. et al. Internal shocks in the jets of radio-loud quasars, *Mon. Not. R. Astron. Soc.* **325**, 1559 (2001).
32. Raiteri C. M. et al. Blazar spectral variability as explained by a twisted inhomogeneous jet, *Nature* **552**, 374-377 (2017).
33. Giannios, D. Reconnection-driven plasmoids in blazars: fast flares on a slow envelope. *Mon. Not. R. Astron. Soc.* **431**, 355-363 (2013).
34. Giannios, D. & Spruit, H. C. The role of kink instability in Poynting-flux dominated jets. *Astron. Astrophys.* **450**, 887-898 (2006).
35. Pushkarev, A. B., Kovalev, Y. Y., Lister, M. L. & Savolainen, T. Jet opening angles and gamma-ray brightness of AGN. *Astron. Astrophys.* **507** L33-L36 (2009)
36. H. E. S. S. Collaboration Constraints on the emission region of 3C 279 during strong flares in 2014 and 2015 through VHE gamma-ray observations with H.E.S.S, 2019arXiv190604996H
37. Böttcher, M.; Els, Paul Gamma-gamma absorption in the broad line region radiation fields of gamma-ray blazars, *Astrophys. J.* **821**, 102B (2016).
38. Shah Z. et al. Study on temporal and spectral behaviour of 3C 279 during 2018 January flare, *Mon. Not. R. Astron. Soc.* **484**, 3168-3179 (2019).

Reviewers' comments:

Reviewer #1 (Remarks to the Author):

I consider that the authors have successfully addressed my concerns.

I recommend that the manuscript is accepted for publication.

Reviewer #1, additional consultation comments (Remarks to the Author):

I went through the arguments of both the referee and the authors.

My impression is that the referee makes some reasonable technical

points that need to be addressed by the authors. On the other hand, concerning the main issue of novelty of the current work, I am inclined to agree with the authors.

The authors clearly identify, in the gamma-ray data, the pattern of fast fares superimposed on a slower envelope and proceed to test a specific model for these flares. If correct, their interpretation has far reaching implications for blazars. It is this combination of data and theoretical interpretation that supports the novelty of this work.

The authors do not claim to be the first to find minute-timescale variability in blazars. So the referee's long list of related publications seems not to be directly relevant.

The Paliya et al. (2015, ApJ, 811, 143) paper comes closer to what the current paper is claiming. I cannot comment for the data analysis in the Paliya et al. work (since it is outside my expertise). Nevertheless, I agree with the authors that the X-ray variability on a timescale comparable to $\sim R_g/c$ that Paliya report is much less constraining/conclusive for the models than the, more extreme, gamma-ray variability seen in 3C 279 (in this case the emitting region also has be located further away putting tighter constraints to all models). As such the current paper makes a clear advancement in the field: it showcases more constraining observations and goes through a more thorough theoretical analysis of the data.

Reviewer #2 (Remarks to the Author):

In the revised manuscript titled "Gamma-ray flares from relativistic magnetic reconnection in the jet of the quasar 3C 279", authors have tried to respond queries. However, I still believe that the manuscript lacks originality in terms of theoretical interpretation of the observation of a characteristic pattern of fast flares superimposed with a slower-varying envelope of day-long emission. The interpretation is already put forth. Also, the observational features are largely reported in previous studies. The authors are encouraged to check the following references.

Minute-scale gamma-ray variability:

1- Meyer, M. et al. 2019, ApJ 877, 39

(this is the best and latest paper reporting the shortest flux variability found in many blazars)

2- Shukla, A. et al. 2018, ApJ, 854, L26

3- Ackermann, M. et al. 2016, 824, L20

4- Aleksic, J. et al. 2011, ApJ, 730, L8

Minute-scale X-ray variability:

1- Zhu, S. F. et al. 2018, ApJ, 853, 34

2- Paliya, V. S. et al. 2015, ApJ, 811, 143

Regarding the pattern of the fast flares superposed on slowly varying events, As I said in earlier report that it has already been reported in Paliya et al. (2015, ApJ, 811, 143). More recently, light curves reported in various papers have shown this kind of features and not explicitly discussed in the context of magnetic reconnection mechanism. A few examples are: Meyer et al. (2019, ApJ, 877, 39), Britto et al. (2016, ApJ, 830, 162), Paliya et al. (2017, ApJ, 844, 32).

The authors have argued that there are data gaps (up to ~40 minutes) in the NuSTAR observation and Paliya et al. (2015) might have mistakenly considered such gaps to be of 5 minute duration. This argument is not correct. The authors are encouraged to go through that paper (see also Foschini et al. 2011, A&A, 530, 77) and check equation 3 which was used to compute the shortest flux doubling time. Paliya et al. (2015) do not report any such constraints as claimed here by the authors about fixing the time interval of two consecutive bins to 5 minutes. In fact, the whole calculation becomes meaningless if one fixes the time difference of two bins ($t-t_0$, see their Eq 3). Furthermore, the luminosity contrast between envelope and flares as seen from the NuSTAR observations is not 1. The authors are encouraged to take a look at other NuSTAR pointings covering the same 2013 April flaring event, where sub-hour scale flux variability is reported and there is a substantial difference between the luminosity of the slowly varying flare and rapid flaring events.

Though the results reported in this paper are worth publishing, the authors cannot argue them as breakthrough findings. Therefore, in my opinion, this article is suitable for more common journals, e.g. ApJ, MNRAS.

Few more comments:

1. In the manuscript, the Fermi analysis is carried with the older version of the IRF "P8R2", however, Fermi data server was updated and new versions "P8R3_V2" are available from 26-Nov-2018. Also, instead of 3FGL catalog, now LAT 8-year Source Catalog (4FGL) is available.

2. If the authors claim that ONLY magnetic reconnection can explain these observations, they also need to address a few relevant questions. For example: what variability pattern is expected at wavelengths other than gamma-rays keeping in mind the magnetic reconnection scenario? Is a correlated flux variability across the electromagnetic spectrum expected and observed during 2018 flare of 3C 279? What about the behavior of the optical polarization (see, e.g., Zhang, H. et al. 2018, ApJ, 862, L25)?

3. In lines 96-97: Furthermore, the detection of a hard spectrum with index $\alpha = 1.69 \pm 0.09$, during the orbit in which minute-scale-flare was detected, strongly indicates particle acceleration through magnetic reconnection. Why other acceleration mechanisms can not explain the harder index. (see recent paper arXiv:1909.04431)

4. Again the location of emission region out-side the BLR region is repeated in several previous works for the source 3C 279. It is not new findings of the manuscript. The authors themselves have quoted this in the lines "Surprisingly, in blazars of the flat-spectrum radio quasar (FSRQ) type exhibiting BLR, the gamma-rays from the compact emission regions must originate light years away from the central black hole: No sign of gamma-ray attenuation due to pair production from collisions between the gamma-rays and the UV photons from the BLR is generally found".

5. Why to consider UV photons, if emission region is located outside BLR region. (lines 154-157)

"Mini-jets from a reconnection region, located where the jet collimation breaks down, produce optically thin gamma-ray emission mainly by the external Compton process and possibly by the photo-production of pions in interactions with UV photons from the BLR and IR photons from the dusty torus."

6. Line 8 in supplementary file: Fermi energy range can go upto 500 GeV

7. Abstract: In the abstract one should avoid more general statements: "When observed under small angles to the line of sight, they are called blazars showing variable non-thermal emission across the electromagnetic spectrum from radio waves to gamma rays".

8. Through out the manuscript the beginning double quote " is reversed.

Response to reviewer's comments: "Gamma-ray flares from relativistic magnetic reconnection in the jet of the quasar 3C 279"

A. Shukla^{1,2*}, K. Mannheim¹

¹*Institut für Theoretische Physik und Astrophysik, Universität Würzburg, Emil-Fischer-Str. 31, 97074 Würzburg, Germany*

²

³*Discipline of Astronomy, Astrophysics and Space Engineering, Indian Institute of Technology, Khandwa Road, Simrol, Indore, India 453552*

We thank both referees for their careful reading of the manuscript and their many constructive suggestions. We have answered all the issues raised by the referees. Reviewer's comments are given in purple color, and the author's reply is provided in black color in the report. All the changes are marked in boldface in the manuscript and also provided in this report with blue text.

*amit008@gmail.com

1 Reviewers' comments:

Reviewer 2 (Remarks to the Author):

Reviewer 2 (Remarks to the Author):

In the revised manuscript titled "Gamma-ray flares from relativistic magnetic reconnection in the jet of the quasar 3C 279", authors have tried to respond queries. However, I still believe that the manuscript lacks originality in terms of theoretical interpretation of the observation of a characteristic pattern of fast flares superimposed with a slower-varying envelope of day-long emission. The interpretation is already put forth. Also, the observational features are largely reported in previous studies. The authors are encouraged to check the following references.

Minute-scale gamma-ray variability:

1- Meyer, M. et al. 2019, ApJ 877, 39 (this is the best and latest paper reporting the shortest flux variability found in many blazars)

2- Shukla, A. et al. 2018, ApJ, 854, L26

3- Ackermann, M. et al. 2016, 824, L20

4- Aleksic, J. et al. 2011, ApJ, 730, L8

Minute-scale X-ray variability:

1- Zhu, S. F. et al. 2018, ApJ, 853, 34

2- Paliya, V. S. et al. 2015, ApJ, 811, 143

Author's reply: It is true that minute scale gamma-ray variability has been observed in

several sources and the papers suggested by the reviewer have discovered it in some different sources. While in some of the quoted publications magnetic dissipation is considered as a viable explanation, here we report for the first time a peak-in-peak variability pattern on minute time scales in coincidence with the discovery of high energy photons above 10 GeV which can be considered key signatures of magnetic reconnection outside of the BLR.

Regarding the pattern of the fast flares superposed on slowly varying events, As I said in earlier report that it has already been reported in Paliya et al. (2015, ApJ, 811, 143). More recently, light curves reported in various papers have shown this kind of features and not explicitly discussed in the context of magnetic reconnection mechanism. A few examples are: Meyer et al. (2019, ApJ, 877, 39), Britto et al. (2016, ApJ, 830, 162), Paliya et al. (2017, ApJ, 844, 32).

Author's reply: While the interpretation of flares of Mkn421 with magnetic reconnection must be applauded, the results of our data analysis of the 2018 giant flare of 3C279 are unique, since no other blazar with a BLR has yet shown comparable giant flares with these signatures. The peak-in-peak variability renders shock acceleration as an explanation very unlikely (the chance coincidence for emission from multiple shocks is low) and the high-energy photons in combination with the presence of the BLR prove dissipation to occur outside of the central lightyear, which may be different in BL Lac-type objects.

In the case of Britto et al. 2016, a few envelope emissions and a separate 20 min flare from the

FSRQ 3C 454.3 were detected in 2014. However, the observed flare activity was not superimposed on the slow-varying envelope emission, which is evident from figure 2 of the Britto et al. 2016. Interestingly, to explain their findings, they have considered the shock-in-jet blazar model instead of invoking the magnetic reconnection model.

Finally, Meyer et al. 2019 have discovered rapid variability in CTA 102 and 3C 279 by analysing ten years of Fermi-LAT data (but not including the 2018 giant flare of 3C279 which we have analyzed). Their analysis does not provide conclusive evidence supporting the magnetic reconnection scenario, but invites further and more rigorous tests of the magnetic reconnection models for consistency, as we performed them.

The authors have argued that there are data gaps (up to 40 minutes) in the NuSTAR observation and Paliya et al. (2015) might have mistakenly considered such gaps to be of 5 minute duration. This argument is not correct. The authors are encouraged go through that paper (see also Foschini et al. 2011, AA, 530, 77) and check equation 3 which was used to compute the shortest flux doubling time. Paliya et al. (2015) do not report any such constraints as claimed here by the authors about fixing the time interval of two consecutive bins to 5 minutes. In fact, the whole calculation becomes meaningless if one fixes the time difference of two bins ($t-t_0$, see their Eq 3). Furthermore, the luminosity contrast between envelope and flares as seen from the NuSTAR observations is not 1. The authors are encouraged to take a look at other NuSTAR pointings covering the same 2013 April flaring event, where sub-hour scale flux variability is reported and there is a substantial difference between the luminosity of the slowly varying flare and rapid flaring events.

Though the results reported in this paper are worth publishing, the authors cannot argue them as breakthrough findings. Therefore, in my opinion, this article is suitable for more common journals, e.g. ApJ, MNRAS.

Author's reply:

The NUSTAR data analysis of Mkn421 reported by Paliya et al. (2015) shows interesting variability patterns and considers magnetic reconnection as a possible interpretation. Since the bulk Lorentz factor is less constrained than in our analysis of 3C 279 data, other interpretations

Figure 1: A light curve obtained from Paliya et al. 2015 shown in the panel "a" for OBS id "60002023025" where ~ 14 min variability is claimed by Paliya et al. on 11 April 2013. The light curve obtained by our analysis (FPMA and FPMB telescope combined) is presented in the panel "b" for the same OBS id "60002023025". The light curve provided in the panel "c" is a standard product of the NuStar FPMB telescope for same OBS id "60002023025". And finally, panel "d" shows the expended view of the light curve between 82.5-100 Ksec for the same OBS id "60002023025". A clear data gap of ~ 40 min is present at 88.125 to 90.625 ks in the light curve, where ~ 14 minute variability is claimed by Paliya et al. We think that authors did not notice this data gap by mistake and considered two adjacent bin time difference to be 5 min, which had resulted in an apparent sub-hour scale (~ 14 min) variability in the Mrk 421.

Figure 2: The luminosity contrast between envelope and flares as seen from the NuSTAR observations, both envelope and flares show similar counts rates.

like shock-in-jet models remain viable as well. Regarding the extraction of variability time scales from the NUSTAR data, we refer the referee to the plots included (Fig. 1 & Fig. 2).

1. In the manuscript, the Fermi analysis is carried with the older version of the IRF "P8R2", however, Fermi data server was updated and new versions "P8R3_V2" are available from 26-Nov-2018. Also, instead of 3FGL catalog, now LAT 8-year Source Catalog (4FGL) is available.

Author's reply: Data used in our publication has been reanalysed considering a 15° ROI, P8R3, IRF: P8R3_SOURCE_V2 using 4FGL catalogue and the old plots have been replaced with new plots in the manuscript. The E_b value using 3FGL was 341 MeV, and it changed to 442 MeV in the 4FGL catalogue. Notably, with this change in E_b value, we have not found any significant difference in our final results except for minor softness in the spectrum.

2. If the authors claim that ONLY magnetic reconnection can explain these observations, they also need to address a few relevant questions. For example: what variability pattern is expected at wavelengths other than gamma-rays keeping in mind the magnetic reconnection scenario? Is a correlated flux variability across the electromagnetic spectrum expected and observed during 2018 flare of 3C 279? What about the behavior of the optical polarization (see, e.g., Zhang, H. et al. 2018, ApJ, 862, L25)?

Author's reply: The steep spectral slopes in the low-energy spectral component and a luminous high-energy EC component are expected if particles are accelerated through magnetic

reconnection (Christie I. M., Petropoulou M., Sironi L., Giannios D., 2019, MNRAS, 482, 65). In this scenario, very sharp cutoffs occur at both low and high frequencies due to the synchrotron-self-absorption process and the Klein-Nishina effect, respectively. Moreover, similar variability patterns of a slow envelope and fast-moving flare are expected, but as the low energy electrons cool slowly, the decays phase of the fast flares and of the envelope will be longer at lower energies. This behaviour is in accord with observations of the 3C279 flare. In general, we expect longer-time-scale radio outbursts to accompany major magnetic dissipation events observed at high energies, but their discussion is beyond the scope of the paper.

Indeed, we expect large-amplitude swings in polarization angle (PA) during reconnection events agreeing with the referee. A significant change is observed in the PA from 3C 279 during the flare over the few days. High-cadence observation of PA are needed to understand the role of PA change in such events during the fast flares. These high-cadence observations were unfortunately not available for the April 2018 flare.

We thank the referee for pointing out the relevance of these features, and we added the information in the supplementary materials in the section §Origin of the fast flare and envelope emission outside of the BLR.

3. In lines 96-97: Furthermore, the detection of a hard spectrum with index $= 1.69 \pm 0.09$, during the orbit in which minute-scale-flare was detected, strongly indicates particle acceleration through magnetic reconnection. Why other acceleration mechanisms can not explain the harder index. (see recent paper arXiv:1909.04431)

Author's reply: We agree with the referee that observed hard spectral index ~ 1.7 in FSRQ can be explained with the help of other emission mechanisms. However, in our case, the detection of peak-in-peak pattern in the light curve, minute scale variability, and the emission of high energy photons from 3C 279, make magnetic reconnection model one of the most viable options. Moreover, the origin of the hard photon spectrum is a natural outcome of magnetic reconnection (Christie I. M., Petropoulou M., Sironi L., Giannios D., 2019, MNRAS, 482, 65). In the case of Lewis et al. 2019 (arXiv:1909.04431), the shortest variability was claimed ~ 3 hours, meaning this flare does not warrant magnetic reconnection at the first place. Therefore the observed hard spectrum can be explained without invoking magnetic reconnection.

4. Again the location of emission region out-side the BLR region is repeated in several previous works for the source 3C 279. It is not new findings of the manuscript. The authors themselves have quoted this in the lines "Surprisingly, in blazars of the flat-spectrum radio quasar (FSRQ) type exhibiting BLR, the gamma-rays from the compact emission regions must originate light years away from the central black hole: No sign of gamma-ray attenuation due to pair production from collisions between the gamma-rays and the UV photons from the BLR is generally found".

Author's reply: We agree that the location of the emission of gamma-ray radiation is constrained to outside of BLR in a few previous works for the source 3C 279. However, these previous works could not distinguish between the shock-in-jet model and the magnetic reconnection model. As we emphasize in our manuscript, only because of the high flux density of 3C 279, we were able

to discover the conjunction of minute-scale variability, peak-in-peak light curve, and absence of pair attenuation as the unique signature of magnetic reconnection outside of the BLR.

Reviewer 2 (Remarks to the Author):

5. Why to consider UV photons, if emission region is located outside BLR region. (lines 154-157) "Mini-jets from a reconnection region, located where the jet collimation breaks down, produce optically thin gamma-ray emission mainly by the external Compton process and possibly by the photo-production of pions in interactions with UV photons from the BLR and IR photons from the dusty torus."

Author's reply: Indeed, the dominating photon field at ~ 0.48 pc in 3C 279 will be due to IR photons from the torus, where the emission zone is located. The BLR UV photon field decreases but still remains present outside of the BLR, still contributing to the mixture of target photon fields (Ghisellini G., Tavecchio F., 2009, MNRAS, 397, 985).

6. Line 8 in supplementary file: Fermi energy range can go upto 500 GeV

Author's reply: Fermi energy range is changed to 500 GeV in the manuscript.

7. Abstract: In the abstract one should avoid more general statements: "When observed under small angles to the line of sight, they are called blazars showing variable non-thermal emission across the electromagnetic spectrum from radio waves to gamma rays"

Author's reply: We changed the wording accordingly to "When the jets are observed at angles of less than a three degrees to the line-of-sight , they are called blazars ..."

8. Through out the manuscript the beginning double quote " is reversed

Author's reply: We have fixed the issue.

Additional changes

1. In the previous manuscript, we noticed that the light curve plotted in Fig. 2-a underwent a forward shift of around eight minutes due to the rounding off of the MJD values when changed to minutes. The same has been corrected in the present manuscript.
2. Both the supplementary data tables have also been revised in the manuscript.
3. All the updated text is provided below:

Text added in the manuscript from line 63-70:

The blazar 3C279 is known for extreme gamma-ray variability²⁵. A giant flare in 2015 showed hourly time scale flux variations occurring outside of the quasar's BLR at a distance of 0.05 parsec estimated for the formation of dissipative shock waves. Now, the analysis of a similar giant flare in 2018 shows minute-scale variability superimposed on the longer duration flare showing that the shock dissipation scenario may not be appropriate for the interpretation, but instead magnetic reconnection might play the dominant role for the dissipation of the jet energy in this stage of its

dynamical evolution.

Text replaced in the manuscript from line 98-100:

A significant (4.7σ) 8-minute (or probably shorter) flare was superimposed on a longer duration envelope at the starting of the first fast flare FF1.

Text replaced in the manuscript from line 101-104:

The average spectrum during this orbit was found to be very hard with photon index $\alpha = 1.74 \pm 0.08$. Fermi-LAT also observed one of the so-far greatest flux from the source (above 100 MeV) during one single orbit $(3.4 \pm 0.23) \times 10^{-5} \text{ ph cm}^{-2} \text{ s}^{-1}$ of the second fast flare of F2.

Text replaced in the manuscript from line 135-138:

Further, by adopting a Lorentz factor of the jet $\Gamma_j \sim 35$, which is required to avoid pair production in the emission zone due to low-energy synchrotron photons and also to avoid a super Eddington jet, and $\epsilon \sim 0.1$, we deduce the size of the reconnection region $l' \sim 8.2 \times 10^{15} \text{ cm}$.

Text replaced in the manuscript from line 148-150:

These values are in good agreement with the observed luminosities of envelope emission $L_{env,obs} \sim 6.7 \times 10^{48} \text{ erg s}^{-1}$ and minute-scale-flare emission $L_{msf,obs} \sim 2.1 \times 10^{49} \text{ erg s}^{-1}$ recorded during flare F2 (cf. detailed calculations in *Methods*).

Text replaced in the manuscript from line 154-156:

we obtained, $R_{diss} \sim 0.48 \text{ pc}$, which also satisfies the $l' \leq R_{diss} \theta_j$ condition which is needed to

keep the reconnection region inside the jet cross section

Text replaced in the Supplementary Methods in th line 10:

20 MeV - 500 GeV

Text replaced in the Supplementary Methods from line 11-20:

DATA_QUAL > 0 && LAT_CONFIG==1 together with the LAT event class == 128 and the LAT event type ==3 was used. Spectral analysis on the resulting data set was carried out by including `gll_iem_v07.fits` and the isotropic diffuse model `iso_P8R3_SOURCE_V2_v1.txt`. The flux and spectrum of 3C 279 were determined by fitting a log parabola model, using a binned `glike` algorithm based on the `NewMinuit` optimiser^{3,4}.

Text replaced in the Supplementary Methods in line 33:

.. the only points having three-sigma detection are plotted.

Text replaced in the Supplementary Methods in line 39:

The hardest average spectrum was found during the third flare with the index $\alpha = 1.87 \pm 0.03$.

Text replaced in the Supplementary Methods in line 62:

... a reduced $\chi^2 \sim 1.8$ with the rise time of the envelope emission $\sim 1.62 \pm 0.06$ d.

Text replaced in the Supplementary Methods from line 63-64:

... a reduced $\chi^2 \sim 1.5$ and rise time for envelope emission of $\sim 1.61 \pm 0.03$ d is obtained.

Text replaced in the Supplementary Methods in line 66:

... a reduced $\chi^2 \sim 1.7$ and the rise time of $\sim 1.47 \pm 0.08$ d.

Text replaced in the Supplementary Methods from line 70-72:

The rise time of the slowly-varying component which acted as an envelope for fast flares had a rise time of $\sim 1.35 \pm 0.34$ d.

Text replaced in the Supplementary Methods from line 84-91:

However, within F2, the source was highly inconsistent with a constant flux during a few orbits with p -values ~ 0.002 , and 0.06 . During the orbit with p -value 0.002 , starting from MJD 58227.90 to MJD 58227.97 shown by a vertical solid grey line in Fig. 1-b, we witnessed a strong few-minute scale variability (flux doubling timescale $\tau \sim 8$ minute) with a significance of 4.7σ . However, when a ten-minute binned light curve is fitted with a constant flux in the same orbit (MJD 58227.90 to MJD 58227.97), we get a p -value $\sim 10^{-5}$. Here, we report the detection of significant (6.6σ) flux doubling timescale of ~ 13 minutes using ten-minute binned light curve.

Text replaced in the Supplementary Methods from line 93-94:

The rise time was found to be 8.6 ± 1.8 minutes with the best fit, yielding a reduced χ^2 of 0.95 .

Text replaced in the Supplementary Methods from line 97-101:

Furthermore, the detection of a hard spectrum with index $\alpha = 1.74 \pm 0.08$, during the orbit in which minute-scale-flare was detected, strongly indicates particle acceleration through magnetic reconnection. Fermi-LAT also observed the one of the greatest flux from the source (above 100 MeV)

during one single orbit $(3.4 \pm 0.2) \times 10^{-5} \text{ ph cm}^{-2} \text{ s}^{-1}$ of second fast flare of F2.

Text replaced in the Supplementary Methods from line 136-137:

... the Lorentz factor of the jet must be around 100,

Text replaced in the Supplementary Methods from line 150-153:

A lower limit of Lorentz factor is obtained by equating the radius of re-confinement shock with the size of the emission zone equivalent to five minutes yielding the Lorentz factor of the jet $\Gamma_j \sim 335$, assuming $\Gamma_j \sim \delta_j$ and $\eta = 0.1$, where δ_j is the Doppler factor of the jet.

Text is added in the Supplementary Methods from line 167-172:

The large-amplitude swings in polarization angle (PA) are expected during reconnection events. A significant change was observed in the PA from 3C 279 during the flare over the few days. Moreover, high-cadence observation of PA are needed to understand the role of PA change in such events during the fast flares. These high-cadence observations were unfortunately not available for the April 2018 flare.

Text replaced in the Supplementary Methods from line 173-175:

The Lorentz factor of a monster plasmoid in the rest frame of the jet γ_p can be estimated using the luminosity contrast between the minute-scale-flare to the envelope emission, $\gamma_p = \delta_p/\Gamma_j \sim 2.2$.

Text replaced in the Supplementary Methods from line 178-180:

Considering $\Gamma_j \sim 35$ which is required to avoid pair production in the emission zone due to low-

energy synchrotron photons and also to avoid a super Eddington jet, ..

Text replaced in the Supplementary Methods from line 183-184:

... a characteristic size $l' = t_{env}\Gamma_j c/(1+z) \sim 8.2 \times 10^{15}$ cm for the magnetic reconnection region in jet's co-moving frame.

Text replaced in the Supplementary Methods from line 185-187:

The rise time of minute-scale-flare produced due to the emission from the monster plasmoid can be expressed as $t_p \sim (1+z)fl'/\delta_p c \sim 493 \times (f/0.1)(l'/8.2 \times 10^{15} \text{ cm})/(\delta_p/85) \text{ s} \sim 8 \text{ min} ..$

Text replaced in the Supplementary Methods in line 195:

$R_{diss} \sim 0.48 \text{ pc}$

Text replaced in the Supplementary Methods from line 198-199:

The isotropic envelope and monster plasmoid luminosities are found to be $L_{env} \sim 7.0 \times 10^{48} \text{ erg s}^{-1}$ and $L_p \sim 2.6 \times 10^{49} \text{ erg s}^{-1}$ using Eq.3 and Eq.4 respectively,

Text replaced in the Supplementary Methods from line 201-203:

Luminosities computed from the jet-in-jet model are in excellent agreement with the observed luminosities of the envelope emission $L_{env,obs} \sim 6.7 \times 10^{48} \text{ erg s}^{-1}$ and the minute-scale-flare emission $L_{msf,obs} \sim 2.1 \times 10^{49} \text{ erg s}^{-1}$ above 100 MeV.

25. Paliya V. S. Fermi-Large Area Telescope Observations of the Exceptional Gamma-Ray Flare from 3C 279 in 2015 June. *Astrophys. J. Lett.* **808**, L48 (2015).

Supplementary Data Table 1: Flare characteristics obtained by fitting a sum of two exponential functions (Eq.1) to all the three flares.

Flare : F1 (58133.0-58139.0) fitted with two Components		
Component [Component #]	T_r (days)	T_d (days)
Envelope [1]	1.62±0.06	1.49±0.05
Fast flare [1]	0.06±0.02	0.04±0.02
Reduced- χ^2 (Degrees of freedom) for F1		= 1.77 (43)
Flare : F2 (58222.0-58232.0) fitted with three Components		
Envelope [1]	1.61±0.03	2.74±0.05
Fast flare [1]	0.05±0.01	0.09±0.02
Fast flare [2]	0.09±0.02	0.08±0.01
Reduced- χ^2 (Degrees of freedom) for F2		= 1.52 (79)
Flare : F3 (58268.0-58276.0) fitted with one Components		
Envelope [1]	1.47±0.08	1.88±0.06
Reduced- χ^2 (Degrees of freedom) for F3		= 1.69 (61)
Flare : F2 (58222.0-58232.0) fitted with eight Components		
Envelope [E1]	1.25±0.17	0.25±0.06
Envelope [E2]	0.42±0.05	0.57±0.05
Envelope [E3]	0.75±0.11	0.46±0.17
Envelope [E4]	0.18±0.07	0.59±0.09
Envelope [E5]	1.35±0.34	1.24±0.08
Fast flare [FF1]	0.06±0.01	0.08±0.01
Fast flare [FF2]	0.09±0.01	0.10±0.01
Fast flare [FF3]	0.11±0.04	0.08±0.03
Reduced- χ^2 (Degrees of freedom) for F2		= 1.14 (65)

3. Cash, W. Parameter estimation in astronomy through application of the likelihood ratio. *Astrophys. J.*, **228**, 939-947 (1979).
4. Mattox, J. R. et al. The likelihood analysis of EGRET data. *Astrophys. J.*, **461** 396 (1996).

Supplementary Data Table 2: Flux and spectral properties of all the three flares observed from 3C 279 during 2018 by fitting a log-parabola model. Here α is the spectral index at E_b which is the scale parameter and β is the curvature parameter.

Epoch	Period MJD	Flux $\times 10^{-6}$ (Ph/cm ² /sec)	α	β	Test Statistics
F1	58133.0-58139.0	13.43 \pm 0.25	2.07 \pm 0.02	0.11 \pm 0.01	12655
F2	58222.0-58232.0	9.57 \pm 0.11	1.97 \pm 0.01	0.1 \pm 0.01	45977
F3	58268.0-58276.0	7.7 \pm 0.20	1.87 \pm 0.03	0.15 \pm 0.02	6649
Envelope [F2]	58222-58227.76 & 58228.40-58232.00	8.53 \pm 0.10	1.99 \pm 0.01	0.10 \pm 0.01	37149
Fast flare [F2]	58227.76-58228.40	24.97 \pm 0.60	1.86 \pm 0.03	0.14 \pm 0.02	16339

REVIEWERS' COMMENTS:

Reviewer #2 (Remarks to the Author):

The Authors have answered most of my quires in the revised manuscript. However, as I pointed in my earlier reports that the manuscript lacks originality in terms of theoretical interpretation of the observation of a characteristic pattern of fast flares superimposed with a slower-varying envelope of day-long emission. In my opinion, the manuscript does not qualify for the publication in Nature Communications.

Response to reviewer's comments: "Gamma-ray flares from relativistic magnetic reconnection in the jet of the quasar 3C 279"

A. Shukla^{1,2*}, K. Mannheim¹

¹*Institut für Theoretische Physik und Astrophysik, Universität Würzburg, Emil-Fischer-Str. 31, 97074 Würzburg, Germany*

²*Discipline of Astronomy, Astrophysics and Space Engineering, Indian Institute of Technology Indore, Khandwa Road, Simrol, Indore, India 453552*

We thank both referees for their careful reading of the manuscript and their many constructive suggestions. Reviewer's comments are given in purple color, and the author's reply is provided in black color in the report.

Reviewer 2 (Remarks to the Author): The Authors have answered most of my queries in the revised manuscript. However, as I pointed in my earlier reports that the manuscript lacks originality in terms of theoretical interpretation of the observation of a characteristic pattern of fast flares superimposed with a slower-varying envelope of day-long emission. In my opinion, the manuscript does not qualify for the publication in Nature Communications.

Author's reply: There are no further comments from the reviewers to be implemented in the final version.

*amit.shukla@iiti.ac.in